# Clone-structured graph representations enable flexible learning and vicarious evaluation of cognitive maps

Dileep George [1✉], Rajeev V. Rikhye[1,2], Nishad Gothoskar [1,3], J. Swaroop Guntupalli [1], Antoine Dedieu[1] & Miguel Lázaro-Gredilla[1]

Cognitive maps are mental representations of spatial and conceptual relationships in an environment, and are critical for flexible behavior. To form these abstract maps, the hippocampus has to learn to separate or merge aliased observations appropriately in different contexts in a manner that enables generalization and efficient planning. Here we propose a specific higher-order graph structure, clone-structured cognitive graph (CSCG), which forms clones of an observation for different contexts as a representation that addresses these problems. CSCGs can be learned efficiently using a probabilistic sequence model that is inherently robust to uncertainty. We show that CSCGs can explain a variety of cognitive map phenomena such as discovering spatial relations from aliased sensations, transitive inference between disjoint episodes, and formation of transferable schemas. Learning different clones for different contexts explains the emergence of splitter cells observed in maze navigation and event-specific responses in lap-running experiments. Moreover, learning and inference dynamics of CSCGs offer a coherent explanation for disparate place cell remapping phenomena. By lifting aliased observations into a hidden space, CSCGs reveal latent modularity useful for hierarchical abstraction and planning. Altogether, CSCG provides a simple unifying framework for understanding hippocampal function, and could be a pathway for forming relational abstractions in artificial intelligence.

[1] Vicarious AI, Union City, CA, USA. [2] Present address: Google, Mountain View, CA, USA. [3] Present address: Massachusetts Institute of Technology, Cambridge, MA, USA. ✉email: dileep@vicarious.com

Vicarious trial and error[1], the ability to evaluate futures by mental time travel, is a hallmark of intelligence. To do this, agents need to learn mental models, or "cognitive maps"[2,3], from a stream of sensory information as they experience the environment around them[4]. Learning these mental abstractions is complicated by the fact that sensory observation is often aliased. Depending on context, identical events could have different interpretations and dissimilar events could mean the same thing[5]. As such, a computational theory for cognitive maps should: (1) propose mechanisms for how context and location-specific representations emerge from aliased sensory or cognitive events, and (2) describe how the representational structure enables consolidation, knowledge transfer, and flexible and hierarchical planning. Most attempts at developing such a theory, which include modeling hippocampus as a memory index, a relational memory space, a rapid event memorizer, and systems-level models of pattern-separation and pattern completion, have not reconciled the diverse functional attributes[6–8] of the hippocampus under a common framework. Recent models have attempted to reconcile the representational properties of place cells and grid cells using successor representation (SR) theory[9–11] and by assuming that these cells are an efficient representation of a graph[12]. However, both these models fall short in describing how flexible planning can take place after learning the environment and are unable to explain several key experimental observations such as place cell remapping in spatial and nonspatial environments[13,14] and the fact that some place cells encode routes toward goals[15,16], while others encode goal values[17,18].

A behaving agent often encounters external situations that look instantaneously similar, but require different action policies based on the context. In these situations, sensory observations should be contextualized into different states. In other times, dissimilar looking sensory observations might need to be merged on to the same state because those contexts all lead to the same outcome. In general, to form a flexible model of the world from sequential observations the agent needs to have a representational structure and a learning algorithm that allows for elastic splitting and merging of contexts as appropriate[5,19]. Moreover, the representational structure should be such that it allows for dynamic planning and handling of uncertainty.

Here we propose a specific higher-order graph—clone-structured cognitive graph (CSCG)—that maps observations onto different "clones" of that observation as a representational structure that addresses these requirements. Using just principles of higher-order sequence learning and probabilistic inference, CSCGs can explain a variety of cognitive map phenomena such as discovering spatial relations from an aliased sensory stream, transitive inference between disjoint episodes of experiences, transferable structural knowledge, and shortcut-finding in novel environments. CSCG's ability to create different clones for different contexts explains the emergence of splitter cells[16], and route-specific encoding[20], which we demonstrate using a variety of experimental settings common in neurophysiology. In a repeated lap-running task[21], CSCGs learn lap-specific neurons, and exhibit event-specific responses robust to maze perturbations, similar to neurophysiological observations. CSCGs can also learn to separate multiple environments that share observations, and then retrieve them based on contextual similarity. Notably, the dynamics of clone-structure learning and inference gives a coherent explanation for the different activity remapping phenomena observed when rats move from one environment to another. By lifting the aliased observations into a hidden space, CSCGs reveal latent modularity that is then used for hierarchical abstraction and planning.

**Clone-structured cognitive graphs as a model of cognitive maps**. The central idea behind CSCGs is dynamic Markov coding[22], which is a method for representing higher-order sequences by splitting, or cloning, observed states. For example, a first-order Markov chain representing the sequence of events $A \rightarrow C \rightarrow E$ and $B \rightarrow C \rightarrow D$ will assign high probability to the sequence $A \rightarrow C \rightarrow D$ (Fig. 1a). In contrast, dynamic Markov coding makes a higher-order model by splitting the state representing event $C$ into multiple copies, one for each incoming connection, and further specializes their outgoing connections through learning. This state cloning mechanism permits a sparse representation of higher-order dependencies, and has been discovered in various domains[22–25]. With cloning, the same bottom-up sensory input is represented by a multitude of states that are copies of each other in their selectivity for the sensory input, but specialized for specific temporal contexts, enabling the efficient storage of a large number of higher-order and stochastic sequences without destructive interference. However, learning dynamic Markov coding is challenging because cloning relies on a greedy heuristic that results in severe suboptimality—sequences that are interspersed with zeroth-order or first-order segments will result in an uncontrolled growth of the cloned states. Although[25] incorporated the cloning idea in a biological learning rule, the lack of a probabilistic model and a coherent global loss function hampered its ability to discover higher-order sequences, and flexibly represent contexts. An effective learning approach should split clones to discover higher-order states, and flexibly merge them when that helps generalization.

Our previous work[26] showed that many of the training shortcomings of dynamic Markov coding can be overcome through cloned hidden Markov models (HMM)—a sparse restriction of an overcomplete HMM[27]. In cloned HMMs, the maximum number of clones per state is allocated up front, which enforces a capacity bottleneck. Learning using the expectation maximization (EM) algorithm figures out how to use this capacity appropriately to split or merge different contexts for efficient use of the clones to represent different contexts. In addition, cloned HMMs represent the cloning mechanism of dynamic Markov coding in a rigorous probabilistic framework that handles noise and uncertainty during learning and inference.

Both HMMs and cloned HMMs assume that the observed data are generated from a hidden process that obeys the Markovian property. That is, the conditional probability distribution of future states, given the present state and all past states, depends only upon the present state and not on any past states. For HMMs, the joint distribution over the observed and hidden states given by the following equation:

$$P(x_1, \dots, x_N, z_1, \dots, z_N) = P(z_1) \prod_{n=1}^{N-1} P(z_{n+1}|z_n) \prod_{n=1}^{N} P(x_n|z_n)$$

(1)

where $P(z_1)$ is the initial hidden state distribution, $P(z_{n+1}|z_n)$ is the probability of transitioning from hidden state $z_n$ to $z_{n+1}$, and $P(x_n|z_n)$ is the probability that observation $x_n$ is generated from the hidden state $z_n$. We assume there are $E$ distinct observations and $H$ distinct hidden states i.e., $x_n$ can take a value from $1, 2, \dots, E$ and $z_n$ can take a value from $1, 2, \dots, H$.

In contrast to HMMs, in the cloned HMMs, many hidden states map deterministically to the same observation (Fig. 1b). The set of hidden states that map to a given observation are referred to as the *clones* of that observation. We use $C(j)$ to refer to the set of clones of observation $j$. The probability of a sequence in a cloned HMM is obtained by marginalizing over the hidden

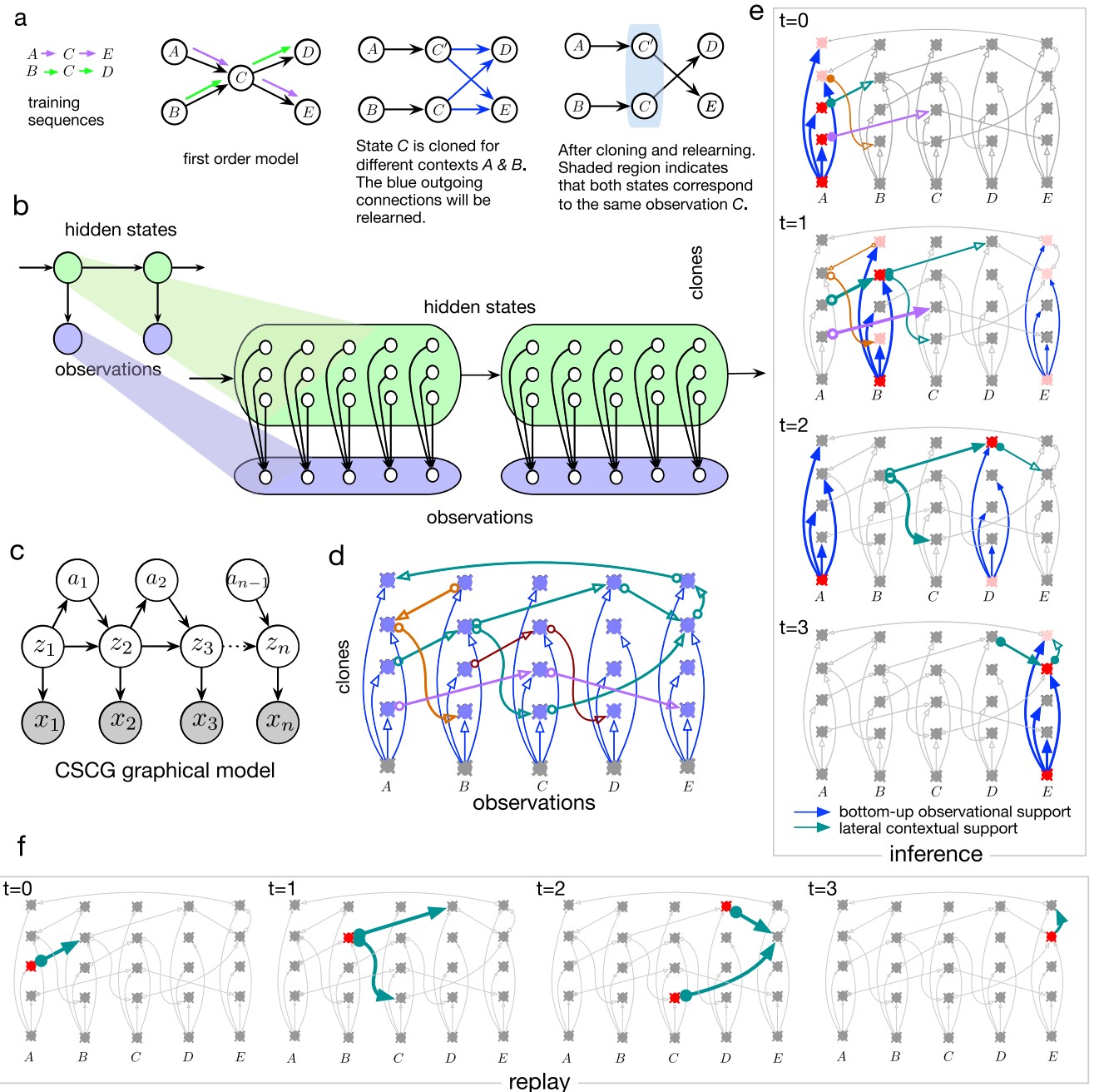

**Fig. 1 Clone-structured cognitive graph. a** Sketch explaining dynamic Markov coding. A first-order Markov chain shown as a graph between nodes representing its states, modeling observation sequences $A \to C \to E$ (purple arrows) and $B \to C \to D$ (green arrows) will also assign high probability to the sequence $A \to C \to D$ because higher-order information is lost at state $C$. (Middle) Higher-order information can be recovered by cloning the state $C$ for different contexts, and then relearning their outgoing connections (blue) to result in the graph on the right. **b** Cloning structure of dynamic Markov coding can be represented in an HMM with a structured emission matrix, the cloned HMM. **c** Probabilistic graphical model for CSCG which extends cloned HMMs in **b** by including actions. **d** Neural implementation of cloned HMM. Arrows are axons, and the lateral connections implement the cloned HMM transition matrix. Different sequences are in different colors, e.g., $A \to C \to E$ in purple. Neurons in a column are clones of each other that receive the bottom-up input (blue arrows) from the same observation. **e** Inference dynamics in the cloned HMM neural circuit. Neural activations strengths are represented in shades of red. Activations that propagate forward are the ones that have contextual (lateral) and observational (bottom-up) support. **f** Replay within the circuit for the sequence $A \to B \to (C, D) \to E \to E$.

states as follows:

$$P(x_1, \ldots, x_N) = \sum_{z_1, \ldots, z_N} P(z_1) \prod_{n=1}^{N-1} P(z_{n+1}|z_n) \prod_{n=1}^{N} P(x_n|z_n)$$

$$= \sum_{z_1 \in C(x_1)} \cdots \sum_{z_N \in C(x_N)} P(z_1) \prod_{n=1}^{N-1} P(z_{n+1}|z_n), \quad (2)$$

where the simplification is a result of $P(x_n = j|z_n = i) = 0$ for all $i \notin C(j)$ (and 1 otherwise). Moreover, since each hidden state is associated with a single observation, EM-based learning is significantly more efficient in cloned HMMs, allowing it to handle very large state spaces compared to standard HMMs[26]. See "Methods" for more details.

A hallmark of our model is the ability to handle noise and uncertainty via message-passing inference[28], and smoothing. Notably, just a forward and backward sweep of messages through the transition matrix $P(z_{n+1}|z_n)$ is adequate for exact inference, and uncertainty about observations is handled through "soft-evidence" messages. Smoothing[29] is a mechanism for incorporating robustness to noise and limited data in probabilistic models. In cloned HMMs, smoothing is accomplished by adding very small probability to some transitions that were unobserved in training. See "Methods" for more details.

**Neurobiological circuit.** Like HMMs[30], cloned HMM can be readily instantiated as a neuronal circuit whose mechanistic interpretation provides additional insights on the advantages of the cloned representation. Each clone corresponds to a neuron, and the "lateral" connections between these neurons form the cloned HMM transition matrix $P(z_{n+1}|z_n)$. For example, the circuit in Fig. 1d shows how neurons can be connected in the cloned HMM to represent the following stored sequences $A \to B \to (C, D) \to E \to A$ (green), $B \to A \to B$ (light brown), $B \to C \to D$ (dark brown), and $A \to C \to E$ (purple).

The transition matrix can also be treated as a directed graph, with the neurons forming the nodes of the graph and the axonal branches forming the directed edges. The set of neurons that are clones of each other receive the same "bottom-up" input (blue arrows) from the observation. The output of a clone-neuron is a weighted sum of its lateral inputs, multiplied by the bottom-up input, corresponding to the forward pass message in HMM inference[30].

The evidence at any particular time instant can be uncertain ("soft evidence"), manifesting as graded activation over the population of observation neurons. For a particular observation, the direct bottom-up connections from the observation to all its clones activate the different sequences that observation is part of, and these activations are then modulated based on the specific contextual support each clone receives on its lateral connections. The population of clone neurons represent the probability of different contexts that are active at any time in proportion to their probability. Figure 1e shows how these activities propagate for a noisy input sequence $A \to (B, E) \to (A, D) \to E$ from $t = 0$ to $t = 3$ corresponding to a true sequence $A \to B \to D \to E$. The activations are represented in different shades of red, with lighter shades indicating weaker activations. At every time instant, the activated lateral inputs are highlighted, and these correspond to the clones active in the previous time step. By correctly integrating the context and noisy input, the clone activations of the cloned HMM filter out the noise to represent the true input sequence. Replay in the hippocampus is the sequential activation of cells that represent prior learning[31]. Replay of previously experienced trajectories is conjectured to be involved in vicarious evaluations of goals[1]. Figure 1f shows how sequences can be replayed (sampled) from the circuit.

Queries like marginal or MAP inference can be implemented in neural circuits as forward and backward sweeps similar to the visualizations in Fig. 1, analogous to the neural implementation of message-passing inference explored in earlier works[28,30,32]. The EM algorithm used for learning is well approximated by the neurobiological mechanism of spike-timing-dependent plasticity[33].

**CSCG: action-augmented cloned HMM.** CSCG extends cloned HMMs to include actions of an agent. An agent's experience is a stream of sensation-action pairs $(x_1, a_1)$, $(x_2, a_2) \ldots (x_{N-1}, a_{N-1})$, $(x_N, -)$ where $x_n \in \mathbb{Z}^*$ are the agent's sensory observations and $a_n \in \mathbb{Z}^*$ are the actions reported by the agent's proprioception.

The observed actions are simply nonnegative integers with unknown semantics (i.e., the agent observes $a_1 = 0$ happened, but does not know that the action means "move north in the room"). In CSCG, the action is a function of the current hidden state and the future hidden state is a function of both the current hidden state and the action taken. The graphical model for this CSCG is depicted in Fig. 1c. Mathematically, the joint observation action density is:

$$P(x_1, \ldots, x_N, a_1, \ldots, a_{N-1}) = \sum_{z_1 \in C(x_1)} \cdots \sum_{z_n \in C(x_n)} P(z_1) \prod_{n=1}^{N-1} P(z_{n+1}, a_n|z_n). \quad (3)$$

Our action-augmented model allows for the agent to learn which actions are feasible in a given state, compared to action-conditioned formulations[34] that only predict future observations from actions.

**Planning within a CSCG.** Planning is treated as inference[35] and achieved using biologically plausible message-passing algorithms[28]. The goal can be specified as either a desired observation or as a specific clone of that observation. Planning is then accomplished by clamping the current clone and the target, and inferring the intermediate sequence of observations and actions required to reach these observations. It is easy to determine how far into the future we have to set our goal by running a forward pass through the graphical model and determining the feasibility of the goal at each step. The backward pass will then return the required sequence of actions. Importantly, because the graphical model is inherently probabilistic, it can handle noisy observations and actions with uncertain outcomes.

## Results

We performed several experiments to test the ability of CSCGs to model cognitive maps. We specifically tested for known functional characteristics such as learning spatial maps from random walks under aliased and disjoint sensory experiences, transferable structural knowledge, finding shortcuts, and supporting hierarchical planning and physiological findings such as remapping of place cells, and route-specific encoding.

**Emergence of spatial maps from aliased sequential observations.** From purely sequential random walk observations that do not uniquely identify locations in space, CSCGs can learn the underlying spatial map, a capability that is similar to people and animals. Figure 2a shows a 2D room with the sensory observations associated with each location. The room has 48 unique locations, but only four unique sensory inputs (represented as colors), and an agent taking a random walk observes a sequence of these sensory inputs. A first-order sequence model would severely under-fit, and pure memorization of sequences will not learn the structure of the room because the same sequence hardly ever repeats. In contrast, a CSCG discovered the underlying 2D

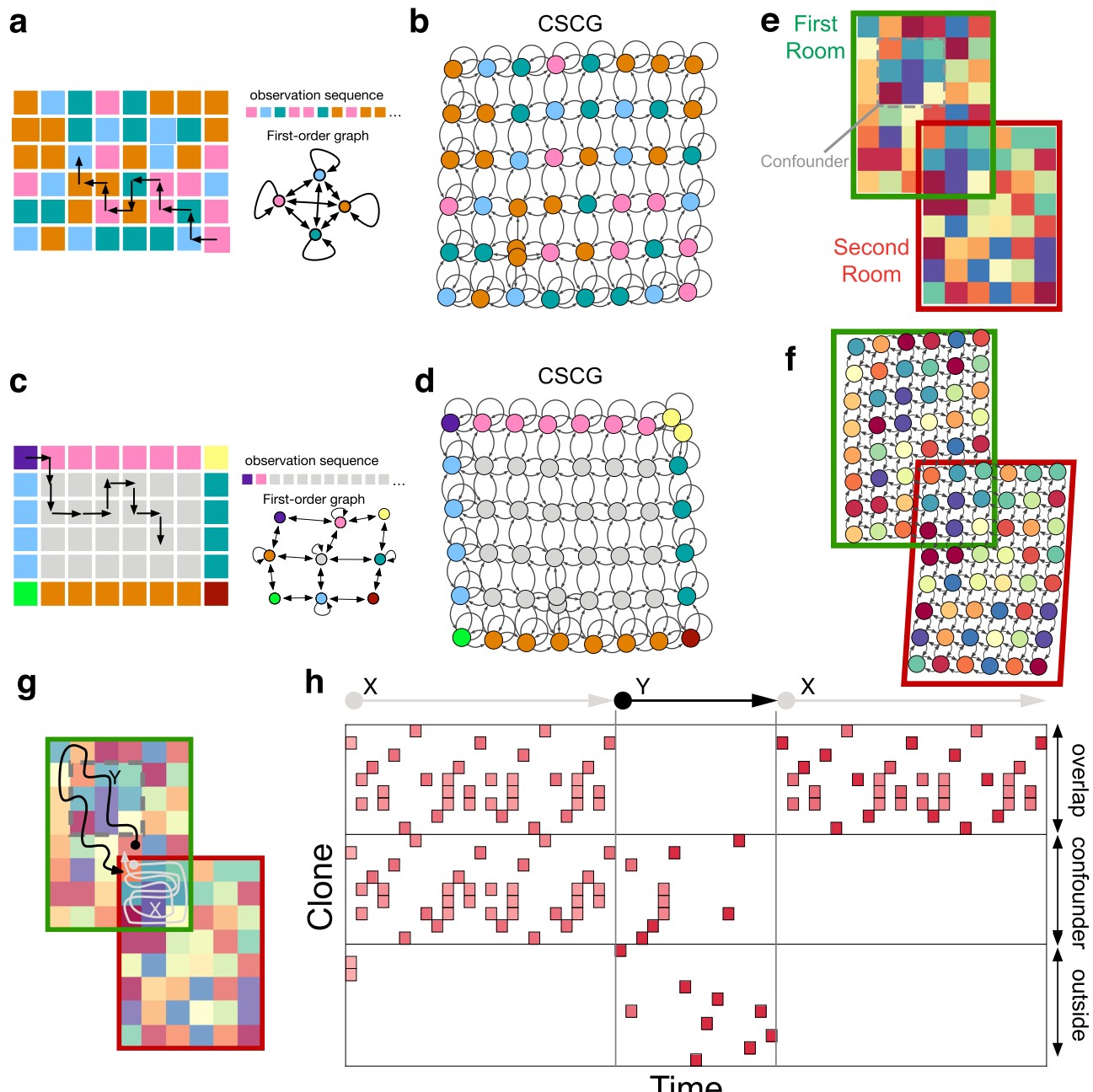

**Fig. 2 Spatial representations emerge from aliased sequential random walk observations without Euclidean assumptions. a** A random walk in a room with only four unique observations (colors) will produce a severely aliased sequence as reflected in the first-order Markov chain. **b** In contrast, transition graph learned by CSCG on random walks in **a** recovers the spatial layout. Nodes in this graph are the clones, and the observation they connect to are indicated by the color of the node. **c** Room with a uniform interior produces aliased sequences highly correlated in time. **d** Transition graph learned by CSCG on random walks in **c**, represented similar to **b**. The redundant yellow nodes (and some brown nodes in **b**) are due to slight imperfections in learning, but do not affect the representation or behavior. **e** An agent experiences two different, but overlapping rooms in disjoint sequential episodes. The overlap region also repeats in the first room, acting as a confounder. **f** As reflected in the transition graph, CSCG performs transitive inference to stitch together the disjoint experience into a coherent global map, and correctly positions the confounder. **g, h** Activation of clones over time as the agent takes the trajectories X (gray), Y (black), and X again in the maze in **g**. Each red square is a clone activation in one time step. During the first traversal of X, the clones corresponding to the overlap and the confounding patch are active because the agent started within the overlap and stayed within. Stepping outside the overlap immediately resolves ambiguity, which is reflected in the clone activity during the traversal Y which includes confounder region and areas outside overlap and confounder, and also during the second traversal of X. See also Supplementary Video 1.

graph of the room perfectly (Fig. 2b), from a sequence of observation action pairs from a random walk with 50,000 steps. As the number of unique randomly placed observations increases, learning becomes easier (see Supplementary Results).

Remarkably, CSCGs learn the spatial topology even when most of the observations are aliased like those from a large empty room where distinct observations are produced only near the walls as shown in Fig. 2c. The combination of high correlation between

observations, and severe aliasing makes this a challenging learning problem. Despite this, the CSCG is able to perfectly learn the topology of the $6 \times 8$ room (Fig. 2d). This capability degrades as the room gets larger, but the degradation is graceful. For example, the periphery of a $9 \times 11$ room is well modeled, but the CSCG is unable to distinguish a few locations in the middle (see Supplementary Results). Moreover, even when the training sequence is purely observations without the paired actions, CSCGs are able to partially learn the layout of the room (see Supplementary Results).

**Transitive inference: disjoint experiences can be stitched together into a coherent whole.** Transitive inference, the ability to infer the relationships between items or events that were not experienced at the same time, is attributed to cognitive maps[7]. Examples include realizing $A > C$ from knowing $A > B$ and $B > C$, or inferring a new way to navigate a city from landmarks and their relative positions experienced on different trips[36].

We tested CSCGs on a challenging problem designed to probe multiple aspects of transitive inference and found that it can stitch together disjoint episodes of sequential experience into a coherent whole. The experimental setting consisted of overlapping rooms (Fig. 2e), each with aliased observations like in the previous experiment. Moreover, the first room had an additional portion which was identical to the overlapping section between the two rooms. This design allows testing whether an agent that experiences only first room or second room exclusively and sequentially can correctly figure out the relationship between the rooms and their overlaps. The combination of a large state-space, aliased observations, nested relationships, and two-dimensional transitivity makes the problem setting significantly harder than previous attempts[37]. We collected two independent 10,000-step sequences of action-observation pairs on each room by performing two separate random walks, and trained a single CSCG on both sequences. The result of training is visualized in Fig. 2f and in Supplementary Movie 2. The learned transition matrix (shown as a graph) has stitched together the compatible region of both rooms, creating a single, larger spatial map that is consistent with both sequences while reusing clones when possible. The confounding additional patch in the first room remains correctly unmerged, and in the right relative position in the first room, despite looking identical to the overlapping region.

Discovering the correct latent global map enables CSCG to make transitive generalizations. Although the agent has never experienced a path taking it from regions that are exclusive to Room 1 to regions exclusive to Room 2, it can use the learned map to vicariously navigate between any two positions in the combined space. Just like in the earlier experiment, the learning is purely relational: no assumptions about Euclidean geometry or 2D or 3D maps are made in the model.

Interestingly, plotting the activation of clones over time reveals that when the agent first traverses the overlapping region (trajectory $X$ in Fig. 2g), clones corresponding to both the overlap region and the identical confounding region are active (Fig. 2h), indicating that the agent is uncertain of its position in the maze. This also suggests that the agent's belief in the cognitive map is split between the two possible realities (see Supplementary Movie 1) because the overlap region and the confounding region are exactly the same without additional context. Stepping out of the overlap region gives the agent adequate context to resolve ambiguity. Subsequently, as the agent explores the confounding region (trajectory $Y$ in Fig. 2g), clones corresponding to this region become more active, and the clones corresponding to the overlap region are no longer active. When the agent returns to the overlap region to follow the same sequence (trajectory $X$) it

originally followed, the clone activities reflect that the agent is no longer confused between the overlap region and the confounding region.

**Learned graphs form a reusable structure to explore similar environments.** The generic spatial structure learned in one room can be utilized as a schema[38,39] for exploring, planning, and finding shortcuts in a novel room, much like the capabilities of hippocampus-based navigation[40]. To test this, we first trained the CSCG on Room 1 based on aliased observations from a random walk with 10,000 steps. As before, CSCG learned the graph of the room perfectly. Next, we placed the agent in Room 2 which is unfamiliar (Fig. 3a). We kept the transition matrix of the CSCG fixed, and re-initialized the emission matrix to random values. As the agent walks in the new room, the emission matrix is updated with the EM algorithm. Even without visiting all the locations in the new room, the CSCG is able to make shortcut travels between visited locations through locations that have never been visited (Fig. 3b). After a short traversal along the periphery as shown in Fig. 3a, we queried to find the shortest path from the end state to the start state. The CSCG returned the correct sequence of actions, even though it obviously cannot predict the observations along the path. Interestingly, Viterbi decoding[41] reveals the same hidden states that you would get if you Viterbi decoded the same path in Room 1. Querying the CSCG on the shortest path from the bottom left corner of the room to the start position reveals the path indicated by the blue arrows in Fig. 3b. This solution is the Djikstra's shortest path through the graph obtained from Room 1. Furthermore, if we "block" the path we get another solution that is also optimal in terms of Djikstra's algorithm (Fig. 3c). Even with partial knowledge of a novel room, an agent can vicariously evaluate the number and types of actions to be taken to reach a destination by reusing CSCG's transition graph from a familiar room.

When the transition matrix from the old room is reused, the new room is learned very quickly even when the agent explores using a random walk: the new room is learned fully when all the locations in the room are visited at least once (Fig. 3d–f). The plots show the proportion of the room explored and the average accuracy of predicting the next symbol as a function of the number of random walk steps.

**Representation of paths and temporal order.** CSCGs learn paths and represent temporal order when the observed statistics demand it, for example when the observations correspond to an animal repeatedly traveling prototypical routes. For example, consider the T-maze shown in Fig. 4a, which is traversed in a figure-of-eight pattern either from the right (blue path) or the left (red path). As a result, the two paths share the same segment. Interestingly, CSCG learns separate clones for this shared segment (Fig. 4b) and similar to the observations in[16], the activity of clones in this overlapping segment will indicate whether the agent is going to turn left or right (Fig. 4c). It is important to note that the ability of CSCGs to learn flexible higher-order sequences is independent of the modality[4]. In particular, the inputs can correspond to spatial observations, odors, sequences of characters, or observations from any other phenomenon[26]. CSCG will learn an approximation of the graph underlying the generative process, in close correspondence with the role for cognitive maps envisaged by[2]. We illustrate in Fig. 4e the CSCG learned for a maze with a shared path shown in Fig. 4d.

Neurophysiological experiments have shown the emergence of "splitter cells" in the hippocampus[16]. These cells represent paths to a goal rather than physical locations and emerge as rats repeatedly traverse the same sequential routes as opposed to

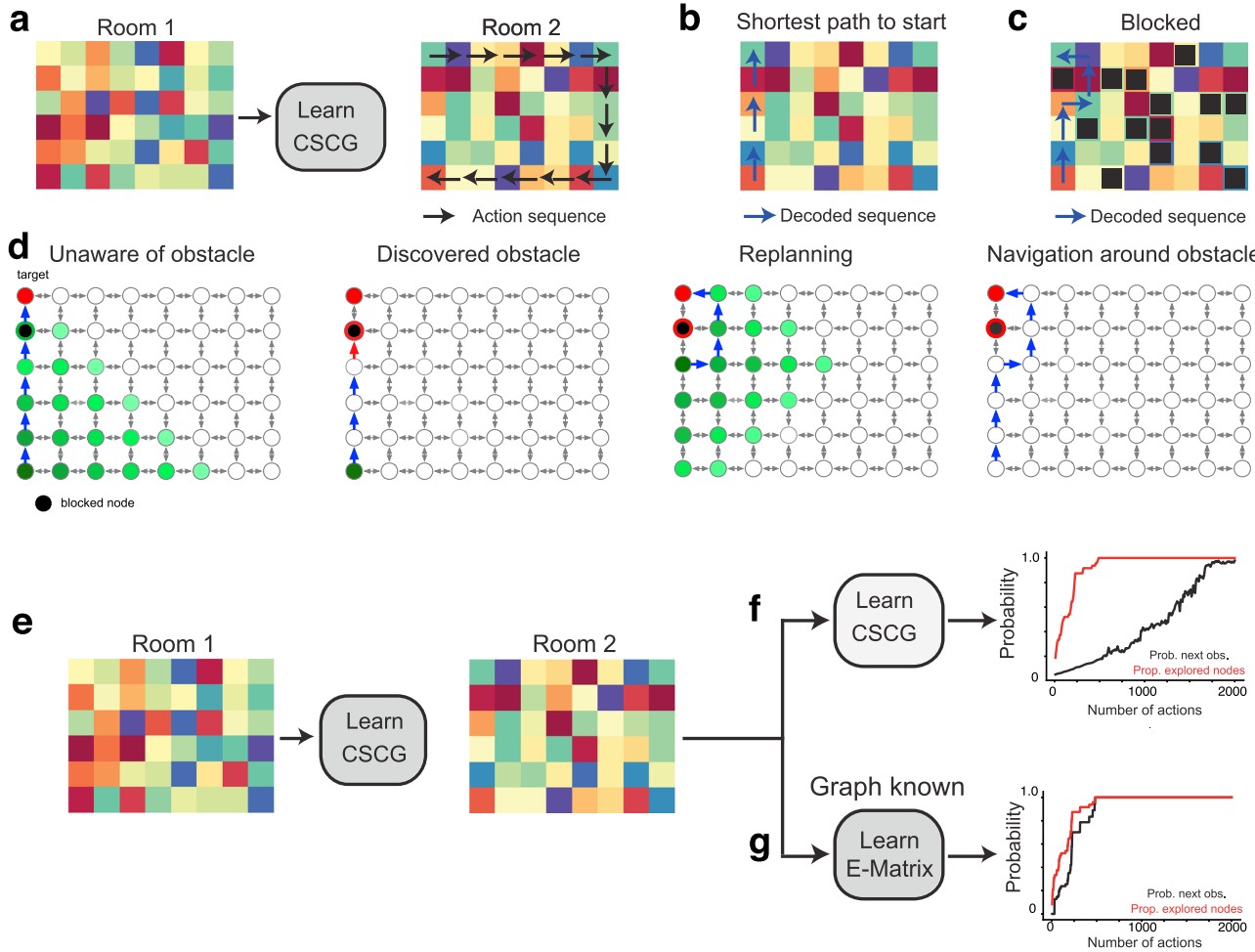

**Fig. 3 Learned transition graphs form a reusable schema.** A CSCG trained on one room (**a**) and partial observations in a second, previously unseen room with identical hidden layout, utilizes the learned structure of the room to rapidly find both the shortest path to the origin (**b**) and navigate around obstacles (**c**). **d** Visualization of message propagation during planning and replanning. Messages propagate outward from the starting clone, and clones that receive the message are indicated in green color. Lighter shades indicate messages that are later in time. The first plan is unaware of the obstacle, and the agent discovers the obstacle only when the action sequence is executed and a planned action fails (red arrow). This initiates a replanning from the new location, and the new plan routes around the obstacle. **e–g** The transition matrix (graph) learned in one room can be used as a reusable structure to quickly learn a new room with the same layout but different observations. Learning is faster when CSCG transition graph is used as a schema to learn the new room (**g**) compared to learning from scratch (**f**).

taking random walks[20]. Figure 4f shows a maze in which the agent can traverse two different routes (indicated by the magenta and green lines) to reach the same destination. Both these routes have regions in which the exact path that the agent follows is stochastic, as denoted by the arrows that indicate the possible movements from each cell. Observations in the maze are marked by numbers and, as before, the same observation can be sensed in many parts of the maze. Additionally, the two routes intersect and share a common segment. CSCGs trained on these paths are able to represent both routes by using different clones for each of the routes, analogous to the route dependency exhibited by place cells in similar experiments. We observe that disjoint subsets of clones will activate when traversing each of the routes. Figure 4g shows that when conditioning on the starting state, sampling in the learned CSCG will always produce paths that are consistent with the two routes. By visualizing the graph defined by the CSCG transition matrix, we see that the two routes are represented with two different chains (Fig. 4g). With a first-order model, when the shared segment is reached, all context about the previous segments will be lost and the model will make incorrect predictions about the future path. CSCGs, on the other hand,

are able to capture the history of the path and therefore properly model the routes and their distinct start states.

Learning higher-order sequences in a CSCG can also explain recently discovered phenomena like chunking cells and event-specific representations (ESR)[21], place cell activations that signal a combination of the location and lap-number for different laps around the same maze. Figure 5a shows a setting similar to the experiment in[21] where a rat runs four laps in a looping rectangular track before receiving a reward. A CSCG exposed to the same sequence learned to distinguish the laps and to predict the reward at the end of the fourth lap, without the help of any lap-boundary markers in the training sequence. Planning for achieving the reward recovered the correct sequence of actions, which we then executed to record the activations of the clones in different laps. Visualizing the propagation of beliefs of each clone, either conditioned on the observation or the action, produces a sequence-like activation pattern where one clone is active for each sensory observation, and as such the different laps around the maze are encoded by different clones (Fig. 5b). Similar to the neurons in the hippocampus, whose firing rates are shown in Fig. 5c[21], clones show graded activity across laps. A clone is

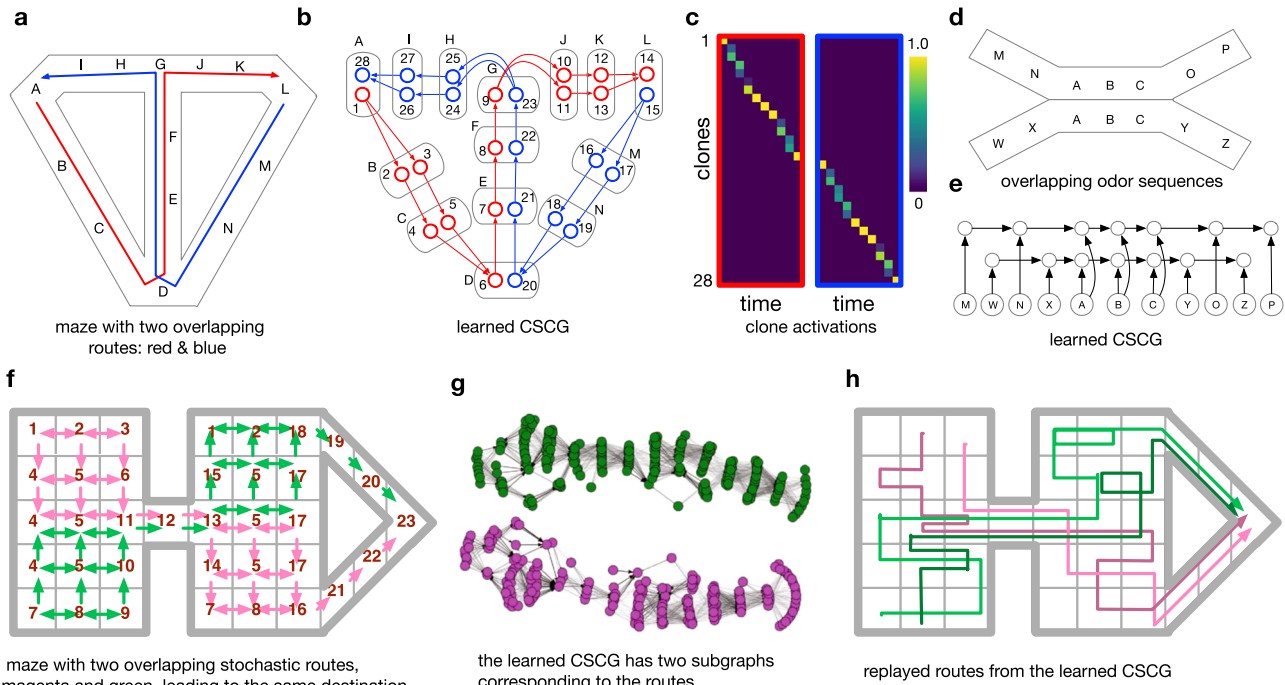

**Fig. 4 Learning temporal order and paths.** In all experiments, CSCG learned the optimal model for prediction, and the learned circuits matched neurobiological observations. **a** Modified T maze from[16] with an overlapping segment between the blue and red paths. A, B, ⋯ , N are the observations at different locations in the maze. **b** CSCG learns route-differentiated clones for the overlapping segment. (The redundant clones on the non-overlapping segments are identical, and due to the learning algorithm not always using the minimal number of clones). **c** Activity of the clones for the right trial, and the left trial. Similar to the observations in[16], the activity of clones in this overlapping segment will indicate whether the agent is going to turn left or right. Distinct neurons are active in the overlapping segment for left-turn trials vs right-turn trials although the observations in the overlapping segment are identical for both trials. Note that clones are not limited to one time step. CSCG learning is able to propagate clones backward into multiple time steps to unravel long overlapping paths. **d** Overlapping odor sequences from[74] **e** Full circuit learned by the CSCG shows that it has learned distinct paths in the overlap, as in[74]. **f** A complex maze in which the agent takes two stochastic paths indicated in magenta and green. Observations in the maze are marked by numbers and, as before, the same observation can be sensed in many parts of the maze. The green and magenta paths overlap in up to seven locations in the middle segment (observations 4-5-11-12-13-5-17). The stochasticity of the paths and the long overlaps make this a challenging learning problem. In contrast to mazes in **a** and **d**, the two paths in this maze lead to the same destination as in[20] **g**. Transition graph learned by the CSCG shows that two different chains are learned for the two routes in **f**, similar to the observation that place cells encode routes, not destinations[20]. **h** Paths replayed from the CSCG after it was trained on sequences from **f**. As they pass through the overlapping segment, the green and magenta routes maintain the higher-order history of where they originated, showing that the learned graph compactly represents the stochasticity and directionality of each route while separating the two routes by appropriately merging and splitting the clones.

maximally active for an observation when it occurs in its specific lap, but shows weak activations when that observation is encountered in other laps, a signature of ESR. This occurs naturally in the CSCG due to smoothing and the dynamics of inference, visualized in Fig. 5e. The cognitive map for this maze is a chain of observations (see Fig. 5e) which split each lap into distinct contextual events. In doing so, the agent is able to identify which lap it is in based on identical local observations. Sun et al. reported that despite extending the maze, neurons in the hippocampus still respond uniquely to each lap. We mimicked this experiment by elongating our maze in one dimension, by introducing repeated, or aliased, sensory observations (Fig. 5d). Again, as with the smaller maze, we observed that clones were uniquely active on each lap and parsed each lap as a separate contextual event (Fig. 5d). Even when the maze is extended by introducing novel observations, the ESR-like clone activity traces persist (see Supplementary Results). Robustness of ESR to maze elongations can also be explained by inference in a smoothed CSCG—a repeated observation is explained as noise in the previous time step, and re-planning from the current observation recovers the correct sequence of actions.

**Learning multiple maps and explaining remapping.** Remapping is the phenomenon where hippocampal place cell activity reorganizes in response to a change in the physical environment. Remapping, which can either be global or partial[19,42–45], depends on how the hippocampus can segregate, store, and retrieve maps for multiple environments that might be similar or dissimilar[13,42].

Similar to the hippocampus[19], a single CSCG can learn to separate maps for different environments that have similar instantaneous observations, represent those maps simultaneously in memory, and then use contextual similarity to retrieve the appropriate map to drive behavior. In Fig. 6a, we show five different 5 × 5 rooms that all share the same 25 observations, but arranged differently in space. We learn a single CSCG from sequences of random walks in each of these rooms where the walks are switched between different rooms at irregular intervals, without providing any supervision about the room identity or time of switching. Although all observations are shared between the rooms, with sufficient training, the CSCG learns to form different clones for the different rooms. Figure 6ai plots the agent's belief about which map it is in as it goes through a 50-step

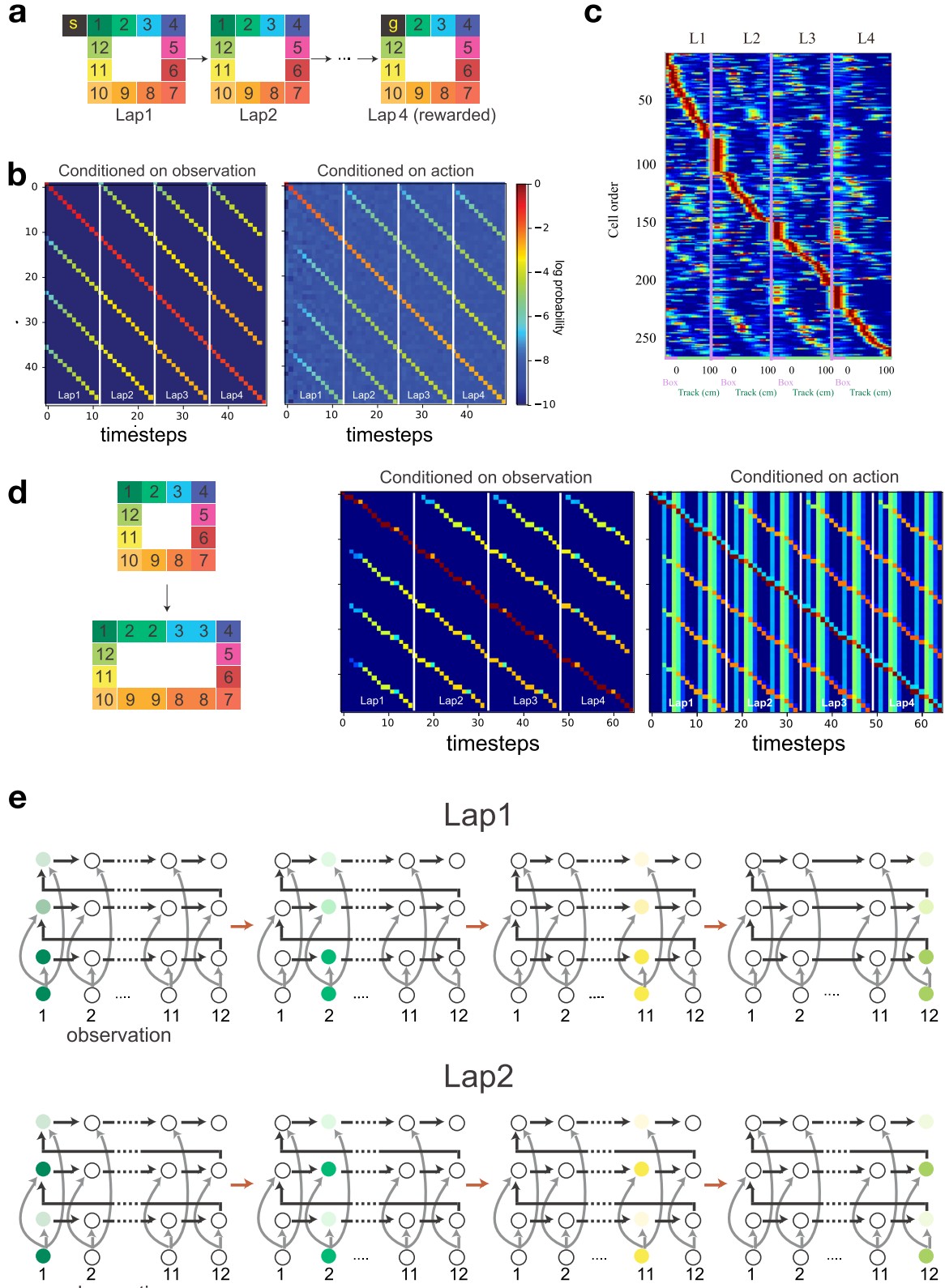

random walk sequence in each room from the first to the last, showing that the maze identity is represented in the population response, despite the ambiguous instantaneous observations.

We conducted a series of experiments to evaluate how the similarity between environments, predictability within each environment, the amount of learning, and the amount of noise and uncertainty affect the degree of reorganization of clone activations. These experiments used two sets of environments—mazes and rooms. Rooms are the 5 × 5 rooms described earlier (Fig. 6a), mazes consist of five different shapes (Fig. 6b) composed of six distinct observations (four different corners, and vertical or horizontal arms). The mazes have better within-environment predictability compared to the rooms because of the lower-branching factor of the random walk, and mazes are more

**Fig. 5 Lap-neurons and event-specific representations. a** A CSCG was trained on observations from four laps around a square maze similar to[21]. The training sequence consisted of one start state, followed by four repetitions of the sequence $1 \rightarrow 2 \rightarrow 3$, .., 12, and then a goal/reward state at the end. It learned to predict the laps perfectly, including the reward at the end of the fourth lap, and planning to get the reward returned the correct sequence of actions. **b** Clone activations (see color map) for the four different laps. Rows correspond to clones. The activations show that there are different clones that are maximally active for different laps, but the other clones are partially active at their corresponding locations, similar to the neurophysiological observations in[21] regarding event-specific-representations. **c** Place cell traces from[21], included with permission. **d** The event-specific representations persist even when the maze is elongated by repeating the observations along the corridor. The CSCG is not trained on the elongated maze. **e** Visualization of the circuit learned by the CSCG including the transition graph, connections from the observations, and activation sequences for laps 1 and 2. The CSCG learned one clone per lap for each position. Smoothing in the CSCG explains why other clones of other laps are partially active. Each row shows how the clone activations transition from observation 1 (left) to observation 12 (right) for the corresponding lap. The active observations are colored in correspondence with **a**, and clone activations are graded in intensity with darker shades being stronger. Overall the visualizations show the circuit dynamics that give rise to the activity traces in **b**.

similar to each other compared to the similarity between different rooms. We trained two CSCGs, one for the set of rooms and the other for the set of mazes, and evaluated how remapping changed with the amount of training, and uncertainty (see Fig. 6ai–iv, bi–iv).

Our observations suggest that global remapping, partial remapping, and rate remapping can be explained using CSCGs: they are manifestations of learning and inference dynamics using a cloned structure when multiple maps are represented in the same model. We were able to reproduce different remapping effects by varying the amount of training and uncertainty. The rows (ii) to (iv) in Fig. 6a, b show the clone activations of the two CSCGs that learned to represent the corresponding rooms and mazes. All the clone activation plots in a column correspond to the same random walk where the agent takes 50 steps in each room/maze, from the first to the last. When the CSCG is fully trained until the EM algorithm converges, the clone activations from the different environments overlap the least, producing an effect similar to global remapping (Fig. 6aiii, biii)[42]. If the CSCGs are partially trained, the clones only partially separate—while many remain exclusive to particular mazes or rooms, a large number are also active in multiple mazes/rooms (Fig. 6aii, bii), corresponding to the effect of partial remapping[13,43]. In a fully trained model, more smoothing, or soft evidence that reflects uncertainty, creates clone activations similar to rate remapping[13,45] (Fig. 6aiv, biv): all the clones that fire in the fully trained setting still fire in this case, but with a lowered rate of firing. This occurs because uncertainty and smoothing causes more sharing of the evidence among clones that represent the same observation.

The similarity between the rooms (mazes), and the amount of predictability within each room (maze), also affects the dynamics of remapping. This can be observed by comparing the clone activity traces for the rooms with that of the mazes in Fig. 6a, b. In Fig. 6bi, the beliefs within each maze are more stable compared to those in the rooms due to the stricter temporal contexts in the mazes[19]. Fluid temporal contexts in the rooms produce more progressive deformation of beliefs[46]. On the other hand, the structural similarity between the different mazes produces more ambiguity at the time of switching, resulting in a longer transient period right after entering a new maze[46]. This is also reflected in Fig. 6bii–iv, where clones in multiple mazes are active right after the point of switching between the mazes.

Taken together, our experiments demonstrate the conditions and mechanisms that determine how the hippocampal network may abruptly switch between preestablished representations or progressively drift from one representation to the other, producing a variety of remapping effects.

**Community detection and hierarchical planning.** Humans represent plans hierarchically[47]. Vicarious evaluations involve

simulating paths to a goal, and hierarchical computations make these simulations tractable by reducing the search space[48]. To enable hierarchical planning, the learning mechanism should be able to recover the underlying hierarchy from sequentially observed data.

By learning a cloned transition graph, CSCG lifts observations into hidden space, enabling the discovery of graph modularity that might not be apparent in the observation. Community detection algorithms[49] can then partition the graph to form hierarchical abstractions[8] useful for planning and inference. Like planning, and inference in CSCGs, community detection can also be implemented using message-passing algorithms[50]. Message-passing algorithms in similar settings are known to have biologically plausible neuronal implementations[51].

We tested CSCGs for their ability to learn hierarchical graphs by simulating the movement of an agent in two mazes. The first maze is a modular graph with three communities where the observations are not unique to a node (Fig. 7a). In earlier studies using this graph[8,10] observations directly identified the nodes, and partitioning the learned CSCG or the SR matrix can reveal the underlying community structure in that fully observed setting (see Supplementary Results). In the current setting of partial observability, due to the degeneracy of observations, community detection or MDS on the SR matrix fails to reveal the hidden communities (Fig. 7b). In contrast, community detection on a CSCG trained from random walks readily reveals the correct community structure. The second maze, shown in Fig. 7d, has a total of 16 rooms arranged as a $4 \times 4$ grid. Each room has aliased observations, and are connected by corridors (black squares). The aliasing is global: instantaneous observations do not identify the room, corridor, or location within a room. Additionally, the maze is structured in such a way that there are four hyper-rooms making this maze a three-level hierarchy. As in the earlier examples, training a CSCG on random walk sequences learned a perfect model of the maze. We then used community detection to cluster the transition matrix of the CSCG (Fig. 7e). This clustering revealed a hierarchical grouping of the clones (Fig. 7f), and a connectivity graph between the discovered communities. The communities respected room boundaries: although some rooms were split into two or three communities, no community straddled rooms. Applying community detection once again on this graph revealed the four hyper-rooms (Fig. 7f) which were the highest level of the hierarchy. To navigate to a particular final destination $F$ from a starting location $S$ using this map, the agent first has to identify in which of these four rooms the goal is located, then plan a route in the community graph between the source community and the destination community (Fig. 7h). In doing so, the search space in the lower level graph is significantly reduced, making planning in the hierarchical CSCG learned graph more efficient than planning directly in the original graph. We implemented this form of hierarchical planning and found

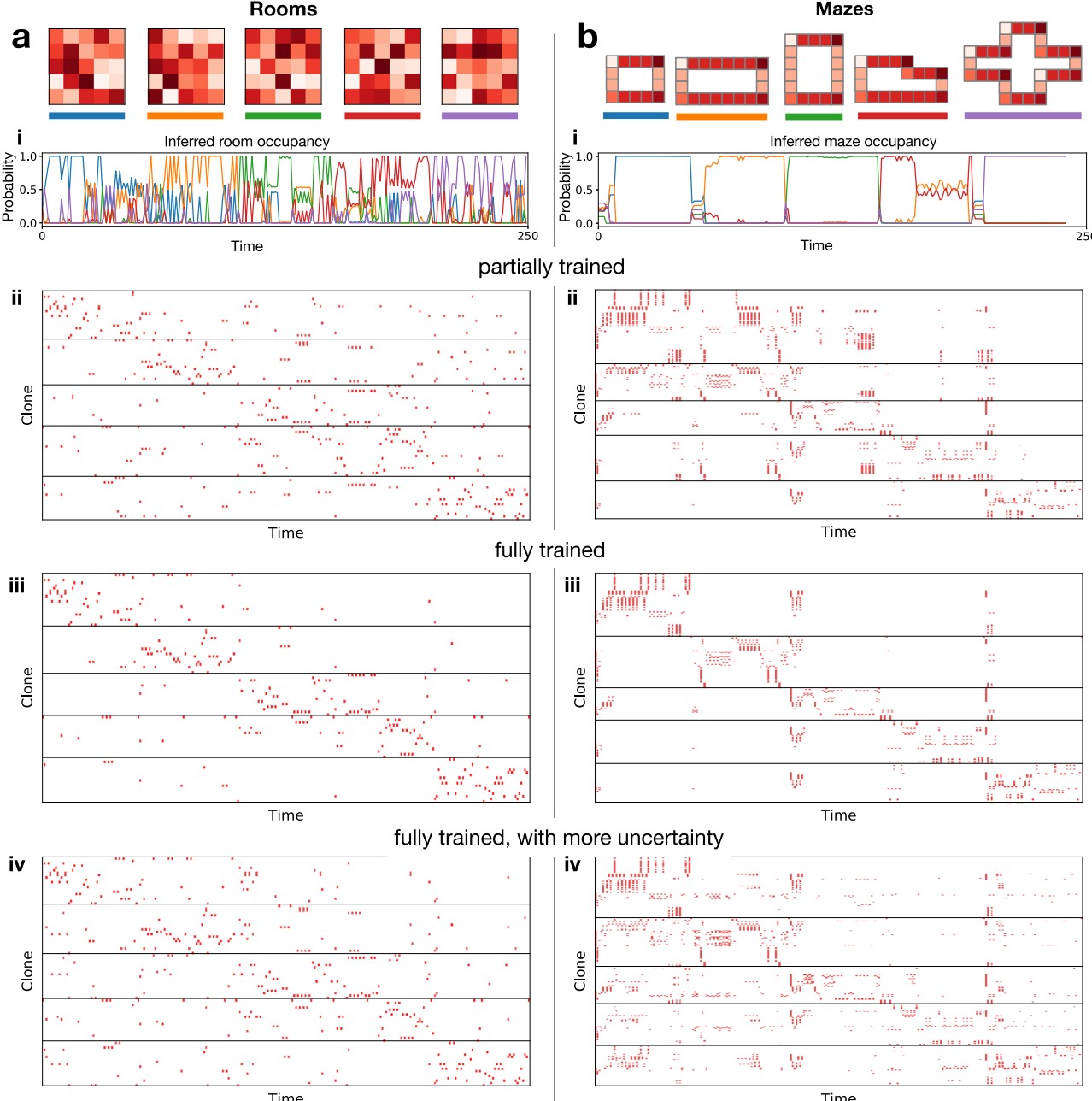

**Fig. 6 Remapping.** Sets consisting of five different rooms (**a**) and mazes (**b**) are used to study activity remapping. In set **a**, the five rooms share 25 different observations, arranged randomly, and in set **b**, the five mazes share six observations arranged in geometrical shapes. Different shades of red represent different observations. Row (i): inferred probability (*Y*-axis) of being in a room/maze as a function of time (*X*-axis). Rows (ii)–(iv) Clone activity traces for a random walk of 50 steps each in rooms (mazes) 1–5 under different conditions (partially trained, fully trained, and more uncertainty). All traces are based on the same random walk and use the same clone ordering. Activity traces corresponds to global remapping in a partially trained CSCG, and partial remapping in a partially trained CSCG. Adding more uncertainty to a fully trained CSCG produces activity traces that correspond to rate remapping.

that we were always able to recover an efficient path between randomly selected start and end position (see Supplementary Methods for more implementation details and for computational efficiency estimates).

Learning higher-order graphs that encode temporal contexts appropriately is crucial for the extraction of the hierarchy using community detection algorithms. Approaches that learn first-order connectivity on the observations, for example, SR on observations[9], will not be able to form the right representations because the observations are typically severely aliased (see Supplementary Fig. 3).

## Discussion
In this paper we pursued the strong hypothesis that the hippo-campus performs a singular sequence learning algorithm that learns a relational, content-agnostic structure, and demonstrated evidence for its validity[4,52]. Realizing this core idea required several interrelated advancements: (1) a learning mechanism to extract higher-order graphs from sequential observations, (2) a storage and representational structure that supports transitivity, (3) efficient context-sensitive and probabilistic retrieval, and (4) and learning of hierarchies that support efficient planning—techniques we developed in this paper. In contrast to approaches

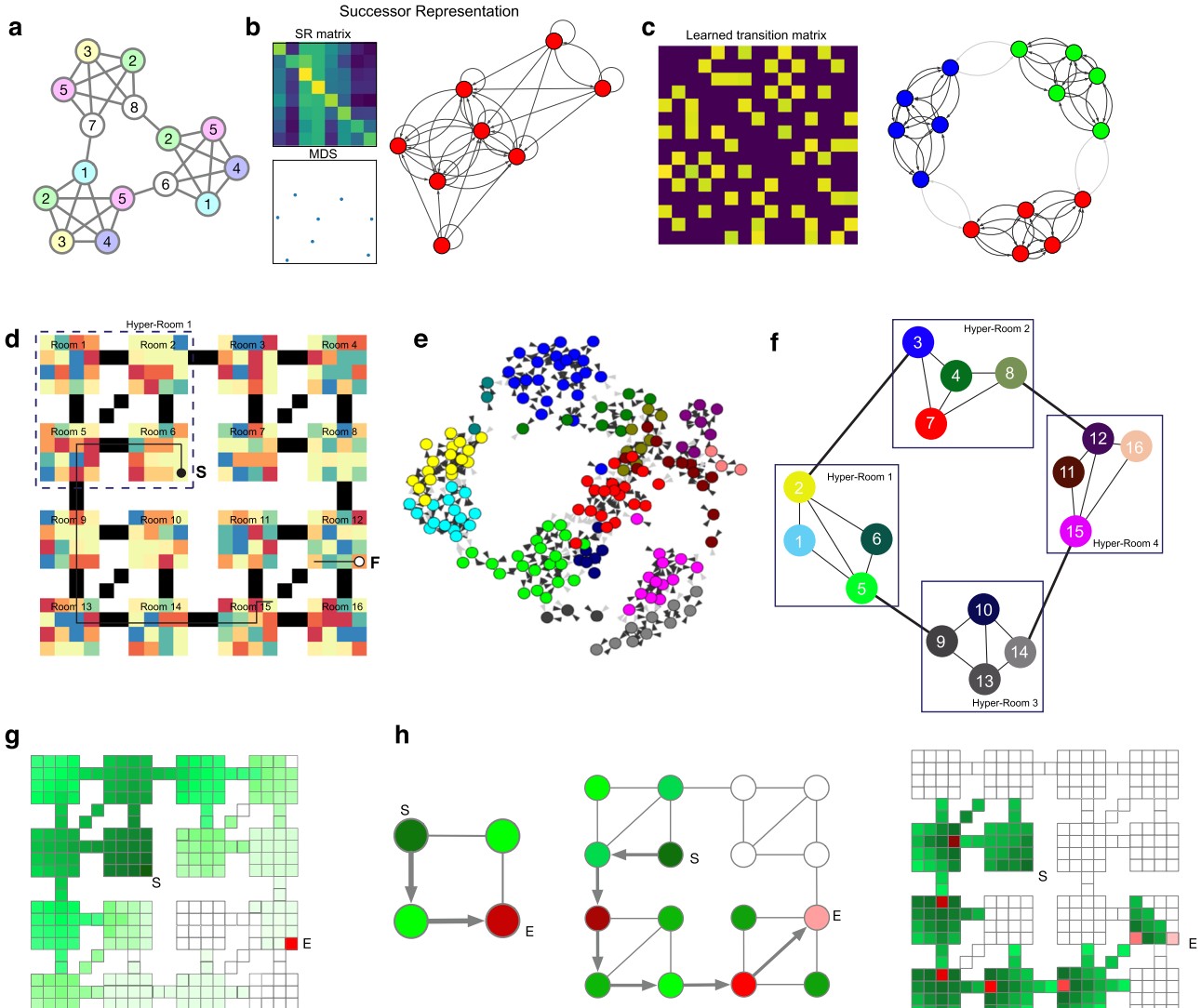

**Fig. 7 CSCGs enable hierarchical abstraction and planning.** The cloned graph of the CSCG lifts the observations into a hidden space, allowing for discovery of modularity that is not apparent in the visible observations. **a** The modular graph from[8], modified to have aliased observations. Observations at each node are indicated by the numbers, and many different nodes produce the same observation. **b** MDS or community detection on the SR matrix of random walks in **a** does not reveal the modularity of the graph. **c** Community detection on the CSCG transition matrix successfully recovers the modularity of graph in **a**, recovering three communities. **d** A maze that has an embedded three-level hierarchy. Sensory observations are aliased both within rooms and across rooms. The black pixels denote "bridges" between the rooms. CSCG is trained on random walks from this maze. Community detection on the learned CSCG transition matrix revealed a first level of organization into rooms (**e**), and another level of community detection revealed hyper-rooms (**f**), resulting in a three-level hierarchical graph reflecting the nested structure of the maze. Planning a path (black arrows) between two rooms (denoted as (S) tart (filled black dot) and (F)inish (open black dot) in **d**) was achieved by finding the shortest path between hyper-rooms to navigate, next finding the shortest path between rooms, and lastly finding the shortest path within the rooms in this reduced search space. **g** Visualization of planning message propagation in the one-level graph. Messages propagate in the whole maze, indicating a wide search area. **h** Visualization of hierarchical planning. Routes are first identified on the highest level, which then becomes sub-goals at the lower level. The red colored nodes indicate the sequence of sub-goals, and their intensities reflect the ordering of the sub-goals. Compared to **g**, hierarchical planning requires fewer messages to be propagated so it is faster.

that model context by concatenating or chunking visible observations, clones in CSCG are latent states that model flexible contexts that can have arbitrarily long temporal dependencies.

As a model CSCG spans multiple levels of the Marr hierarchy —its computational specification is based on probabilistic models and optimal inference, its algorithmic realization utilized neuroscience insights[23] and readily translates to a neurobiological implementation that offers mechanistic explanations for all the experimental phenomena we considered. In a true biological implementation of CSCG, a clone state might be represented by a small assembly of neurons, and that would not change the underlying representation. The core representation learning

mechanisms of CSCG might be implemented in areas CA3 or CA1 of the hippocampus[5], with CSCG-based decision making implemented in the orbitofrontal cortex[53].

CSCGs share similar motivations with other hippocampal theories like Tolman Eichenbaum machine (TEM)[34,54], and SR, but differ substantially in capabilities and tradeoffs. For instance, unlike TEM, CSCGs can plan to achieve arbitrary goals selected at test time (see Fig. 3b, c) and natively handle uncertainty, and erroneous or ambiguous observations (see Retrieval and Remapping in the Supplementary material). CSCGs also allow for efficient exact inference, which enables sophisticated queries to be answered quickly and exactly. In contrast, the representational

complexity of TEM only allows for approximate inference. SR[9,10,55] represents the current state of an agent by aggregating distributions over its future locations for a given policy. While the policy dependence of SR has a computational advantage for a fixed reward distribution, dynamic planning requires recomputing the SR[10,56]. In contrast, CSCG can maintain higher-order temporal dependency and allows for dynamic planning with message-passing. Although SR can be used to find communities, it requires the world to be fully observable. By lifting degenerate observations into a latent graph, CSCGs can reveal latent hierarchies, a capability that is yet to be demonstrated in TEM or SR. The observation of grid cell-like properties in the eigenvectors of the SR could be a property of all methods that employ a transition matrix (see Supplementary Results), and we suspect that this property in itself might not have any behavioral relevance.

In concordance with[52], CSCGs represent sequences of content-free pointers: each pointer can be referring to a conjunction of sensory events from different modalities. These pointers themselves can be clustered according to a similarity metric[57], and the cluster centers will then be the atoms that are sequenced by CSCG. Partial matches of a multi-dimensional conjunction could partially activate multiple clusters and be treated as soft evidence by the CSCG, providing a potential explanation for multi-modality of place cells in various situations[14]. The output from grid cells is treated as just another sensory modality. By providing a periodic tiling of uniform space, grid cell outputs could help to learn maps when other sensory cues are degenerate.

Although beyond the scope of the current work, hippocampal replay[31] could potentially be explained using CSCGs. Replay has two distinct roles in CSCG. First, post-learning replay is used for consolidating trajectories using Viterbi training (see "Methods"). Our related work[58] has shown that an algorithm that rapidly memorizes and gradually generalizes is possible for learning a CSCG representation, and the gradual generalization step uses replay for consolidation. Second, behavior-time replay is used in CSCG for searching of trajectories to multiple goals and their vicarious evaluations.

CSCGs have intriguing connections to schema networks[59], and to schema-like representations in the hippocampus[38]. Creating different clones for different temporal contexts is similar to the idea of synthetic items used to address state aliasing[60,61] in relational representations. In addition, since sequence learning takes place in many other brain areas, for example the parietal cortex[62] and the orbitofrontal cortex[63], a natural extension of this work would involve learning higher-order conceptual relationships and applying them to cognitive flexibility. Similarly, encoding snapshots from a graphical model for vision[64,65] as the input to this sequencer might enable the learning of visuo-spatial concepts and visual routines[66], and model the bi-directional influence hippocampus has on the visual cortex[67]. We believe these ideas are promising paths for future exploration.

The present work can be further extended by combining it with the active inference framework[68] which provides a guiding principle for combining exploration and exploitation. Intuitively, each E-step of the EM algorithm updates posterior beliefs about hidden states and corresponds to state estimation or inference. Conversely, the M-step updates point estimators of model parameters and can be construed as learning. As with the TEM, these expectation maximization-based schemes (e.g., the Baum–Welch algorithm) effectively ignore uncertainty about the parameters and replace posteriors over parameters with point estimates or Delta functions. This has the computational benefit of not having to worry about conditional dependencies between posterior densities over states and parameters. Conversely, in active inference approximate posteriors are updated over both states and parameters. This means that uncertainty about parameters

nuances estimates of hidden states and vice versa. Technically, active inference schemes, in this setting, generally use some form of variational Bayes under a mean-field approximation (of which the EM algorithm can be seen as a special case). The mean-field approximation for the parameters of the transmission and emission matrices are generally parameterized in terms of Dirichlet distributions, which leads to simple update schemes that include both the uncertainty about model parameters and their expected values. Although not considered here, including uncertainty about parameters can be important in terms of optimizing exploration or searches—to minimize uncertainty about emission (i.e., likelihood) and transmission (i.e., prior) probabilities. Active inference and CSCGs adopt a probabilistic formulation that accommodates uncertainty using hierarchical priors over model parameters. This offers an avenue for further research into structure learning and planning as inference, in this setting. For example, see[69] for an application of active inference to spatial planning, navigation, and path cells.

Elucidating how cognitive maps are represented in the hippocampus, how they are acquired from a stream of experiences, and how to utilize them for prediction and planning is not only crucial to understand the inner workings of the brain, but also offers key insights into developing agents with artificial general intelligence. The CSCG model, which we introduce in this paper, provides a plausible answer to each of these questions. We expect this model to be beneficial in both neuroscience and artificial intelligence as a way to produce explicit representations that are easy to interpret and manipulate from multimodal sequential data.

## Methods

**Expectation maximization learning of cloned HMMs.** The standard algorithm to train HMMs is the EM algorithm[70] which in this is context is known as the Baum–Welch algorithm. Cloned HMM equations require a few simple modifications with respect the HMM equations: the sparsity of the emission matrix can be exploited to only use small blocks of the transition matrix both in the E and M steps and the actions, if present, should be grouped with the next hidden state (see Fig. 1c), to remove the loops and create a chain that is amenable to exact inference.

Learning a cloned HMM requires optimizing the vector of prior probabilities $\pi$: $\pi_u = P(z_1 = u)$ and the transition matrix $T$: $T_{uv} = P(z_{n+1} = v | z_n = u)$. To this end, we assume the hidden states are indexed such that all the clones of the first emission appear first, all the clones of the second emission appear next, etc. Let $E$ be the total number of emitted symbols. The transition matrix $T$ can then be broken down into smaller submatrices $T(i, j)$, $i, j \in 1 \ldots E$. The submatrix $T(i, j)$ contains the transition probabilities $P(z_{n+1} | z_n)$ for $z_n \in C(i)$ and $z_{n+1} \in C(j)$ (where $C(i)$ and $C(j)$, respectively, correspond to the hidden states (clones) of emissions $i$ and $j$).

The standard Baum–Welch equations can then be expressed in a simpler form in the case of cloned HMM. The E-step recursively computes the forward and backward probabilities and then updates the posterior probabilities. The M-step updates the transition matrix via row normalization.

**E-step**

$$\alpha(1) = \pi(x_1) \quad \alpha(n+1)^\top = \alpha(n)^\top T(x_n, x_{n+1})$$
$$\beta(N) = 1(x_N) \quad \beta(n) = T(x_n, x_{n+1})\beta(n+1)$$

$$\xi_{ij}(n) = \frac{\alpha(n) \circ T(i, j) \circ \beta(n+1)^\top}{\alpha(n)^\top T(i, j)\beta(n+1)}$$
$$\gamma(n) = \frac{\alpha(n) \circ \beta(n)}{\alpha(n)^\top \beta(n)}.$$

**M-step**

$$\pi(x_1) = \gamma(1)$$
$$T(i, j) = \left(\sum_{n=1}^{N} \xi_{ij}(n)\right) \oslash \left(\sum_{j=1}^{E} \sum_{n=1}^{N} \xi_{ij}(n)\right).$$

where $\circ$ and $\oslash$ denote the element-wise product and division, respectively (with broadcasting where needed). All vectors are $M \times 1$ column vectors, where $M$ is the number of clones per emission. We use a constant number of clones per emission for simplicity here, but the number of clones can be selected independently per emission.

*Computational savings.* For a standard HMM with $H$ hidden states, the computational cost for running one EM step on a sequence of length $N$ is $\mathcal{O}(H^2 N)$ and the required memory is $\mathcal{O}(H^2 + HN)$ (for the transition matrix and forward-backward messages). In contrast, a cloned HMM exploits the sparse emission matrix: with $M$ clones per emission, the computational cost is $\mathcal{O}(M^2 N)$ and the memory requirement is $\mathcal{O}(H^2 + MN)$, in the worst case. Also, there will be additional savings for every pair of symbols that never appear consecutively in the training sequence (since the corresponding submatrix of the transition matrix does not need to be stored). Memory requirements can be improved further by using the online version of EM described in the Supplementary materials.

Since $H = ME$, where $E$ is the total number of symbols, an increase in alphabet size will increase the computation cost of HMMs, but will not affect the cost of cloned HMMs.

Intuitively, the computation advantage of cloned HMMs over HMMs comes from the sparse emission matrix structure. The sparsity pattern allows cloned HMMs to only consider a smaller submatrix of the transition matrix when performing training updates and inference, while HMMs must consider the entire transition matrix.

*CSCG: action-augmented cloned HMM.* CSCGs are an extension of cloned HMMs in which an action happens at every time step (conditional on the current hidden state) and the hidden state of the next time step depends not only on the current hidden state, but also on the current action. The probability density function is given by Eq. (3), and reproduced here for convenience

$$P(x_1, \ldots, x_N, a_1, \ldots, a_{N-1}) = \sum_{z_1 \in C(x_1)} \cdots \sum_{z_n \in C(x_n)} P(z_1) \prod_{n=1}^{N-1} P(z_{n+1}, a_n | z_n),$$

and the standard cloned HMM can be recovered by integrating out the actions. All the previous considerations about cloned HMMs apply to CSCGs and the EM equations for learning them are also very similar:

**E-step:**

$$\alpha(1) = \pi(x_1)\alpha(n+1)^\top = \alpha(n)^\top T(x_n, a_n, x_{n+1})$$
$$\beta(N) = 1(x_N)\beta(n) = T(x_n, a_n, x_{n+1})\beta(n+1)$$

$$\xi_{ikj}(n) = \frac{\alpha(n) \circ T(i, a_n, j) \circ \beta(n+1)^\top}{\alpha(n)^\top T(i, a_n, j)\beta(n+1)}$$
$$\gamma(n) = \frac{\alpha(n) \circ \beta(n)}{\alpha(n)^\top \beta(n)}.$$

**M-step:**

$$\pi(x_1) = \gamma(1)$$

$$T(i, k, j) = \sum_{n=1}^{N} \xi_{ikj}(n) \oslash \sum_{k=1}^{N_a} \sum_{j=1}^{E} \sum_{n=1}^{N} \xi_{ikj}(n).$$

where $N_a$ is the number of actions and $T(i, k, j) = P(z_{n+1}, a_n = k | z_n)$ for $z_n \in C(i)$ and $z_{n+1} \in C(j)$, i.e., a portion of the action-augmented transition matrix.

*Smoothing.* We have observed that convergence can be improved by using a small pseudocount $\kappa$. A pseudocount is simply a small constant that is added to the accumulated counts statistic matrix $\sum_{n=1}^{N} \xi_{ikj}(n)$ and ensures that any transition under any action has non-zero probability. This ensures that at test time the model does not have zero probability for any observations stream. When the pseudocount is only used to improve convergence, one can run EM a second time with no pseudocount, warmstarting from the result of the EM with pseudocount. To use the pseudocount, we only need to change our transition matrix update to be $T(i, k, j) = (\kappa + \sum_{n=1}^{N} \xi_{ikj}(n)) \oslash \sum_{k=1}^{N_a} \sum_{j=1}^{E} (\kappa + \sum_{n=1}^{N} \xi_{ikj}(n))$. The pseudocount can be interpreted as the hyperparameter of a Laplacian prior that is set on the transition matrix, and EM as solving MAP inference for such hyperparameter. As any prior, the pseudocount has a regularization effect that helps generalization when the amount of training data is small in comparison with the capacity of the model.

It might seem at first as if adding a pseudocount would destroy the block-sparse property of the transition and therefore some of the aforementioned computational advantages of the CSCG. However, it is easy to see that the resulting transition matrix can still be expressed as the sum of a block-sparse matrix (with the same sparsity pattern as before) and a rank-1 matrix (which is not stored explicitly, but as the two vectors whose outer product produce it). By doing this, the pseudocount can be used without increasing the computational complexity or the storage requirements of any of our algorithms (EM learning, inference, etc.).

*Inference.* Since the resulting model (with the action $a_n$ and hidden state $z_{n+1}$ collapsed in a single variable) forms a chain, inference on it using belief propagation (BP) is exact. When no evidence is available for a given variable, BP will simply integrate it out, so we can for instance train a model with actions and then, at test time, use it even if no actions are available. We can still ask the model which observation is the most likely in the next time step, or even several time steps ahead, and BP will produce the exact answer by analytically integrating over all

possible past and future actions, and even over the unseen future observations when necessary.

The same model can be used to generate sequences (e.g., to generate plausible observations and actions that would correspond to wandering in a previously learned room) simply by applying ancestral sampling[71] to the conditionals that describe the model after learning (i.e., the transition and emission matrices).

A consequence of the above for spatial data is that an agent roaming the world can infer where in an environment it is located ($z_n$) and then predict which actions are feasible at that location, which is useful for navigation. One can even condition on a future location to discover which set of actions can take you there, and which observations you are expected to see on the way there, see e.g., Fig. 3b, c. This is essentially planning as inference[35].

All of this flexible querying is performed by running a single algorithm (BP) on the same model (without retraining) and only changing the selection of which evidence is available and which probabilistic predictions are requested.

## Experimental details

*Emergence of spatial maps from aliased sequential observations.* For this experiment we collected a stream of 50,000 action-observations pairs. We learned a CSCG with 20 clones (a total of 360 states) with pseudocount $2 \cdot 10^{-3}$ and ran EM for 1000 iterations. This gets a result that is very close to the global minimum: when Viterbi decoded, only 48 distinct states are in use, which is the theoretical optimum on a $6 \times 8$ grid. Viterbi training[41] is used to refine the previous solution.

*Transitive inference: disjoint experiences can be stitched together into a coherent whole.* Figure 2e showcases the CSCG's ability to stitch together two disjoint room experiences when the rooms overlap. For this experiment, we randomly generate two square rooms of size $8 \times 6$ with 15 different observations each. We make both rooms share a $3 \times 3$ patch in their corners as shown in Fig. 2e.

We sample a random walk of length 10,000 of action-observation pairs on each room, always avoiding to take actions that would make the random walk move outside of the room. We use 20 clones, which is enough to fully recover both rooms separately, and use a pseudocount of $10^{-2}$. We run EM (on both sequences simultaneously, as two independent observations of the same CSCG) for a maximum of 100 iterations. After EM convergence, we additionally use Viterbi decoding (with no pseudocount) to remove unused clones. The learned CSCG is visualized in Fig. 2f, showing that the two rooms that were experienced separately have been stitched together. Predictive performance on the stitching of the two rooms is perfect (indicating that learning succeeded) after a few observations required for the agent to locate itself. Notice that there is another patch in the first room that is identical to the merged patches, but was not merged. The model is using the sequential information to effectively identify patches that can be merged while respecting the observational data and context, and not simply looking for locally identical patches to merge.

*Learned spatial maps form a reusable structure to explore similar environments.* For this experiment, we train on a $6 \times 8$ room using 10,000 action-observation pairs. We call this Room 1, see Fig. 3a. There are only 20 unique symbols in the room, some of which are repeated. The pseudocount is set to $10^{-2}$ and we use 20 clones (in this case, only seven clones are strictly required to memorize the room). The regularizing pressure of the pseudocount effectively removes redundant clones. Training is done using EM for a maximum of a 100 iterations. This results in an almost perfect discovery of the underlying graph. Then we set the pseudocount to 0 and continue the training using Viterbi training[41]. This results in perfect discovery of the underlying graph with no duplicate clones. Predictions become perfect after a few initial observations required to know where in the room we are.

When we try to partially learn Room 2 with a few samples from its periphery (see Fig. 3a), we create a new CSCG with the transition matrix that we learned from Room 1 and keep it fixed. The emission matrix is initialized uniformly and learned using EM. The whole data for learning the emission matrix are only the 20 action-observation pairs seen in Fig. 3a. At that point, we fix the model and query it for a return path plan, both with and without blockers in the path. The results are displayed in Fig. 3b, c.

In Fig. 3e–g, we showcase the increase in data efficiency when we transfer the learned topology to a new room with different observations. First, we ignore the results from training on Room 1 and train on a new room, Room 2, from scratch following the same procedure outlined above. We train on the first $N$ action-observation pairs and predict for the rest. We average (geometrically) the probability of getting the next observation right for the last 8000 samples of the 10,000 available. This results in the graph in Fig. 3e, where $N$ is shown in the horizontal axis. Then, we repeat the same procedure, but instead of training from scratch from a random transition matrix with fixed emissions, we fix the transition matrix that we got from training in Room 1 and we learn the emission matrix, which is initialized to uniform. EM for the emission matrix converges in a few iterations. Once all nodes have been observed (when the red curve achieves 1.0), this procedure converges to perfect predictions in one or two EM iterations. This results in the graph in Fig. 3f, where again the horizontal axis shows $N$, the number of training action-observation pairs.

*Representation of paths and temporal order.* To learn the CSCG on the maze in Fig. 4f, we sample 5000 paths along each of the stochastic routes that are shown. The number in each square indicates the observation received at that location, and the arrows indicate possible transitions. We consider both sequences as independently generated by the model, and run EM to optimize the probability of both simultaneously. We allocate 20 clones for each other observation. By inspecting the sum-product forward messages of BP at each step as the agent navigates the two routes, we can see the distributions over clones. We observe they are over disjoint subsets of the clones. To generate the paths from the CSCG shown in Fig. 4g (producing only paths that are consistent with the route), we sample an observation from the normalized messages from hidden state to observation during forward message passing. Finally, to extract the communities and generate the visualization in Fig. 4g, we run the InfoMap algorithm[72] on the graph defined by CSCG transition matrix.

In Fig. 5, we replicate the experiments of Sun et al.[21] as follows. First, we learned a CSCG with 20 clones per observation on a sequence of observations sampled from four laps around the maze shown in Fig. 5d. The start and end positions were unique observations. Training was terminated when perfect learning of the underlying graph was achieved. Community detection (explained below) revealed that each sensory observation was encoded by a unique clone, akin to the chunking cells found by Sun et al.

*Retrieval and remapping.* We generate random walks (random actions out of up, right, down, left) of length 10,000 in each of five mazes. For Fig. 6a, the mazes are $5 \times 5$ rooms where the observations are assigned to cells by a random permutation of the values 1–25, inclusive. For Fig. 6b, the structure of the mazes is shown and the observations are indicated by the color of the cells. We constructed these mazes such that have many shared observations, but each has some distinct structure that differentiates it from the others.

For each of the experiments, we learn a CSCG on these random walk sequences. After learning, we sum the forward messages of sum-product BP in each maze to get a distribution over hidden states for each maze. Now on a test sequence, we can use the forward messages and these clone distributions per maze to infer the probability of being in each maze at each time step. In each of subfigure of Fig. 6, we shows these predictions as well as the distribution over clones over time.

Learning a CSCG in these maze environments can also enable error correction of noisy/corrupted observations. To correct errors in a corrupted observation sequence we modify the emission matrix to generate a random symbol with a small probability, thus modeling errors. Then we perform sum-product message passing on sequences with errors and find the most likely a posteriori value for each symbol. In our case, we only perform a forward pass, which provides an online estimation (based only on past data) of the MAP solution. We will use a corruption probability of 20% in our experiments, uniform over the incorrect symbols. For the $5 \times 5$ rooms, this procedure was able to correct 50 of the 55 corrupted symbols while not corrupting any of the uncorrupted symbols. For the mazes, this procedure was able to correct 46 of the 54 corrupted symbols while, again, not corrupting any of the uncorrupted symbols.

*Community detection and hierarchical planning.* In all figures, custom Python scripts were used to convert the transition matrix of the CSCG into a directed graph. This graph was then visualized using built-in functions in `python-igraph` (https://igraph.org/python/). Similarly, community detection was performed using igraph's built-in `infomap` function.

In the hierarchical planning experiments shown in Fig. 7, we first generated each room by drawing a random integer between 1 and 12 with repetitions to serve as observations. The rooms were then connected via bridges (observation 13, colored black) and were tiled to form a maze as shown in Fig. 7d. Next, we trained a CSCG with 40 clones per observation using 1000 random restarts as described above. The learned CSCG achieved perfect prediction accuracy, suggesting perfect learning of the underlying graph. Community detection on the learned CSCG was performed using igraph. To form the top-level graph shown in Fig. 7e, we collapsed each distinct community into a single node. These communities roughly corresponded to each of the rooms in the maze. In some cases, certain rooms were partitioned into multiple communities. Next, we ran community detection on this graph to retrieve the hyper-rooms.

To compute the shortest trajectory between two locations on the maze, we first computed the shortest path in the highest-level graph using Dijkstra's algorithm, implemented in `networkx` (https://networkx.github.io/). This returned the sequence of hyper-rooms and rooms to be visited in order to reach the goal from a start point. Next, we pruned the community graph to include only clones corresponding to these rooms and then found the shortest path in this reduced graph, which gave the exact sequence of observations from the start position to the goal (denoted by the black arrow in Fig. 7d). This hierarchical approach was consistently better than searching for the shortest path on the full maze itself with an average 25% fewer steps ($n = 10$ mazes). To determine to what extent the partitioning of the CSCG transition matrix into communities helped planning, we formed surrogate communities which no longer respected room boundaries. This resulted in a planned trajectory with an average 35% more steps than the hierarchical plan.

It is important to note that hierarchical planning is significantly more efficient. A representative example of the reduction in complexity can be given as follows. Assume that we have $V$ communities, $E$ inter-community edges, and $M$ nodes inside each community. Further, assume that the nodes inside each community are fully connected, and between any two communities, there is at most one edge. Then with hierarchical planning the complexity of running Djikstra's algorithm is $\mathcal{O}(E + V\log V + N_t(M(M-1)/2 + M\log M))$, where $N_t$ is the number of top-level nodes to traverse in the second planning stage. In contrast, on the full graph, the complexity is $\mathcal{O}(E + MV\log MV + V(M(M-1)/2))$. From these equations, it is easy to see that hierarchical planning is more efficient because $N_t \leq V$ in all graphs.

**Adaptive and online EM variant of CSCGs.** Although the sequences in the experiments of this work are not too large, we might want to be able deal with cases in which there is a very long incoming stream of observations, so long that we cannot even store it in its entirety. In order to handle this case, we can simply extend the previous EM algorithms to make them online.

The adaptive, online version of the EM algorithm in the "Methods" section is obtained by splitting the sequence in $B$ batches $b = 1 \ldots B$ and performing EM steps on each batch successively. This allows the model to adapt to changes in the statistics if those happen over time. The statistics $\xi_{ij}(n)$ of batch $b$ are now computed from the E-step over that batch, using the transition matrix $T^{(b-1)}$ from the previous batch. After processing batch $b$, we store our running statistic in $A^{(b)}$ as:

$$A_{ij}^{(b)} = (1-\lambda)\sum_{k=1}^{b} \lambda^{b-k} \sum_{n \in \text{batch}(k)} \xi_{ij}(n)$$

and then compute the transition matrix $T^{(b)}$ as:

$$T^{(b)}(i,j) = A_{ij}^{(b)} \oslash \sum_{j=1}^{E} A_{ij}^{(b)}$$

where $0 < \lambda < 1$ is a memory parameter and $n \in \text{batch}(k)$ refers to the time steps contained in batch $k$. For $\lambda \to 1$, $T^{(b)}(i,j)$ coincides with the transition matrix from the "Methods" section. For smaller values of $\lambda$, the expected counts are weighed using an exponential window (Normalization of the exponential window is unnecessary, since it will cancel when computing $T^{(b)}(i,j)$.), thus giving more weight to the more recent counts.

To learn from arbitrarily long sequences, we consider an online formulation and express $A_{ij}^{(b)}$ recursively:

$$A_{ij}^{(b)} = \lambda A_{ij}^{(b-1)} + (1-\lambda)\sum_{n \in \text{batch}(b)} \xi_{ij}(n),$$

so that the expected counts of the last observed batch are incorporated into the running statistics.

**Reporting summary.** Further information on research design is available in the Nature Research Reporting Summary linked to this article.

## Data availability

All generated data are available at https://github.com/vicariousinc/naturecomm_cscg.

## Code availability

All simulation code and plotting scripts[73] are available at https://github.com/vicariousinc/naturecomm_cscg.

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

## Acknowledgements

We thank Chen Sun (MIT) for helpful discussions and the use of Fig. 5c. We thank members of Vicarious AI for critically reading this manuscript and for insightful discussions. This work was partially supported by grant N00014-19-1-2368 from the Office of Naval Research.

## Author contributions

D.G.: conceptualization, methodology, running experiments, software, visualization, structuring the paper, writing original draft, response to reviews and editing, funding acquisition, and supervision. R.V.R.: running experiments, software, visualization, writing original draft, response to reviews, and editing. N.G.: running experiments, software, visualization, writing sections. J.S.G.: contributing ideas, review, and editing. A.D.: software, running experiments, and writing sections. M.L.-G: conceptualization, formal analysis, methodology, running experiments, writing sections, visualization, response to reviews and editing, funding acquisition, and supervision.

## Competing interests

The authors declare no competing interests.
