## [Peer Review File · Nature Communications]

Reviewers' Comments:

Reviewer #2:

Remarks to the Author:

Many thanks for responding to my previous points. I think this manuscript is much improved. I appreciate that you have had to deal with lots of other reviewer comments; however, some of my key suggestions still need to be addressed. Perhaps you could focus on the following three:

1. Original recommendation: If you want to bring yourself up to speed with active inference in this setting, I would recommend ... (Kaplan and Friston, 2018) ... with many points at convergence with your own work. ... there is some pressure on you to consider the active inference literature in this area – and explain to readers where the different schemes sit in relation to each other.

Your response: We thank the reviewer for this very helpful suggestion. We have included several references that the reviewer suggests and juxtapose our model with active inference in the discussion section of our paper.

Very good. However, you do not cite or discuss Kaplan, R. and K. J. Friston (2018). "Planning and navigation as active inference." *Biol Cybern* 112(4): 323-343. Can I suggest you look at this paper and replace the following sentence in your discussion:

"Although active inference has been so far used in far simpler models that would not be able to solve the experiments presented in the current work, CSCG's probabilistic formulation is compatible with representing the certainty of the model using hierarchical priors over model parameters, offering an avenue for future research."

with

"Active inference and CSCG's adopt a probabilistic formulation that accommodates uncertainty using hierarchical priors over model parameters. This offers an avenue for further research into structure learning and planning as inference, in this setting. For example, please see (Kaplan & Friston, 2018) for an application of active inference to spatial planning, navigation, and path cells."

2. Original recommendation: I think it would be useful to include the following here: "Intuitively, each step of the EM algorithm updates posterior beliefs about hidden states and corresponds to state estimation or inference. Conversely, the M step updates point estimators of model parameters and can be construed as learning. As with the Tolman Eichenbaum machine, these expectation maximisation-based schemes (e.g., the Baum-Welsh algorithm) effectively ignore uncertainty about the parameters and replace posteriors over parameters with point estimates or Delta functions. This has the computational benefit of not having to worry about conditional dependencies between posterior densities over states and parameters."

Your response: We thank the reviewer for this suggestion and have included this paragraph in the

manuscript.

You have not included this paragraph. Please include it. Although technical, this will be useful for people who understand the maths.

3. Original recommendation: you could then explain why this differs from active inference with something like:

"Conversely, in active inference approximate posteriors are updated over both states and parameters. This means that uncertainty about parameters nuances estimates of hidden states and vice versa. Technically, active inference schemes, in this setting, generally use some form of variational Bayes under a mean field approximation (of which the EM algorithm can be seen as a special case). The mean field approximation for the parameters of the transmission and emission matrices are generally parameterised in terms of Dirichlet distributions, which leads to simple update schemes that include both the uncertainty about model parameters and their expected values. Although not considered here, including uncertainty about parameters can be important in terms of optimising exploration or searches – to minimise uncertainty about emission (i.e., likelihood) and transmission (i.e., prior) probabilities."

Your response: We thank the reviewer for this very helpful suggestion. We have included this paragraph in the discussion section of our revised manuscript.

You have not included this paragraph. Please include in it. Although technical, this will be useful for people who understand the maths.

Reviewer #4:

Remarks to the Author:

Review: „Learning cognitive maps as structured graphs for vicarious evaluation“

Summary

The manuscript discusses on a mechanistic level how cognitive maps could be represented and how this framework possibly could apply to cell activities of the hippocampal formation. The presented algorithm enables multiple representations ("clones") of identical sensory observations in different environments/contexts while maintaining exact spatial/relational information. Interestingly, such a "cloned" representation then allows for behavioral responses specific to a current context. The authors also propose a neurobiological-circuit on how these cloned observations could be represented on a cellular level, for instance, how recruited cell activity on a population level could signal the context of a given environment and handle uncertainty. The link to (cognitive) neuroscience is drawn through a vast number of classical neurophysiological experiments that are adapted to be solved by the proposed algorithm. Beside some interesting similarities between the model-driven results, the algorithm also would allow investigating further questions.

In sum this is an exciting manuscript using a novel computational approach. However, (biological) plausibility of the model is not clear to me (see below) and the correction of some formal errors would significantly improve the manuscript and convey the main idea more clearly.

Major Points

Dimensionality / Higher-Order Graph / Complexity Reduction

Reduction of dimensionality or compression of low-level representations is useful when you want to use/represent data more efficiently - this seems especially plausible when an abstraction of latent state spaces or environments enables fast and simple inference or computations. Therefore, such detailed action-observation representation also containing plenty of redundant information (see Figure 2c and d) might not be the most (biologically) plausible nor efficient form. Yet it enables extracting all exact locations in an environment, even if observations are aliased.

The authors explain how a first-order graph would "under-fit the sequence/data". How does this compare to the CSCG? It appears that in contrast, the CSCG is partly over-fitting a sequence by combining even redundant/aliased observations with unique actions (Fig. 2d), resulting in this 1:1 representation with all exact spatial locations/relations.

Could such representation also be learned without any action-observation pairs only resulting from a sequence/stream of sensory observations without actions or knowledge about relations that caused each transition?

High Dimensionality

The authors should comment on how their model but also the proposed neural circuit could represent multidimensional environments and potentially extract only what is salient and/or relevant at different settings. For instance, stimuli/observations containing information from multiple dimensions, where only two dimensions are relevant to solve a task. Or somewhat more generally spoken, experiences in the real life are never exactly the same, therefore the question of how the model would differentiate relevant and irrelevant information and could represent only the most necessary/abstract information.

(Neuro-)biological plausibility

The proposed neural circuit idea is quite compelling and interesting. As proposed, the output of neurons (for one observation) is weighted by the sum of its lateral inputs but also multiplied by the bottom-up input (Line 123). This could also have some great applications regarding predictive coding. As described in Line 139 the "[...] population of clone neurons represent the probability of different contexts [...]" this mechanism could also be quite convincing on how to handle noisy environments/input. And how "soft evidence" in case of uncertainty could be passed-on through lateral inputs.

But a few questions remain unclear. Could the authors comment on how such single neuron representations might emerge in the first place? And additionally, argue how efficient this one neuron one representation/clone mechanism might be. Potentially regarding other cells such as place cells, where single neurons represent/respond to multiple environments (see also Remapping). Or even the same place and grid cells reportedly represent both spatial and non-spatial information also across sensory modalities (Aronov et al. 2017). In contrast, in their proposed neural circuit one single cell only would respond to one observation from one specific context, if aliased this would result in another cellular representation. This might lead to a large quantity of neurons representing complex/high-dimensional environments or non-spatial dimensions.

Context

Regarding the context (encoded from the weighted lateral input of neurons, Line 131), I wondered whether the authors could specify what comes first: Single-cells that encode a current context vs. a novel context/environment that leads to aliased and new single-cell representations? Maybe the authors could speculate what other structures and input would be needed to form novel single-cell representations of aliased observations or a novel environment/context. For instance, it is suggested that the hippocampus determines when to create a new state or update an old one (see Niv 2019), and vice versa, the context is needed to detect novelty. Moving away from the mechanistic approach/single-cell representations, the authors should also comment how different brain areas might be relevant for the acquisition and representation of different cognitive maps, e.g., abstract task states that are represented in the OFC (Schuck & Niv 2019).

Known Graph Structure (Figure 3)

Since the graph/room layout stays exactly the same between relearning of a novel arrangement of observations (room 1 vs. room 2, I wondered how the re-learning performance of only partly overlapping or even incongruent layouts with aliased observations would look like – if applicable please provide data on that.

Lap Counting Task (Figure 5)

In the methods section, the start and stop for each lap of Fig. 5a task is described as unique (Line 789: “The start and end positions were unique observations.”). Could you please clarify if these start and end positions were unique for each lap vs. only for the first and last lap of the task. If yes, this manipulation seems unfortunately not comparable to the original study by Sun et al. (2020). Adding a unique observation to each start of a lap is changing the initial task substantially. Therefore, the observed results are less surprising and less comparable to the cellular response reported by Sun et al. (2020).

A unique observation between the laps is clearly allowing the CSCG to differentiate the different laps and even be able to count the unique observations (laps) until a specific goal is reached (4 laps). In contrast, in the original paper by Sun et al. (2020), all observations were the same for the rodents navigating a maze for 4 laps until a reward was received. Albeit all laps were exactly the same, yet the cellular pattern showed this distinctive result included in figure 5c. Since this task and results are quite important for their manuscript in explaining the plausibility and also capabilities of the proposed model, the authors should clarify this task detail and your findings regarding this slight but important change of the task manipulation.

Additionally, one of the many manipulations in Sun et al. (2020) was to elongate the trained maze, as similarly proposed in this manuscript (Figure caption 5.d. – unfortunately not part of the methods section). Here I would also argue against the comparability of both tasks. In your manuscript „ [...] 4 laps of 12 steps each [...] “(Line 407) were elongated but with a fixed/the same amount of observations (i.e., 12). How would the event-specific representation in Fig. 5.d. change, if also the number of observations would be increased (elongated maze -> more observations)? Furthermore, it would be interesting to see, how these representations would change through a different shape of the maze or after training on 4 laps but showing the reward (unique observation/goal) after 5 laps, etc. See other manipulations in Sun et al. (2020).

Remapping (Figure 6)

Regarding re-mapping (Fig 6.), I would recommend highlighting, that only one CSCG was trained on all different rooms (and one for the maze set). Since all rooms were aliased this results in representing different rooms with different clones for each observation. Here it is interesting to see, how a different amount of training or noise results in differences in the beliefs and identifying the rooms. But in respect of the proposed neural circuit (biological plausibility) cloned observations (of different rooms) would be represented by different neurons and this, in fact, is not entirely in accordance with the common assumption of re-mapping of e.g., place cell activity.

Also, please be considerate with the “cause or result” of remapping, whether CSCG produces remapping or could explain remapping. Please also check for inconsistencies in the text and figure captions, since this could be misleading.

Hierarchical Abstraction (Figure 7)

As reported in Fig 7 a-b) the SR fails to reveal modularity in the given task with aliased observations. To be fair or more transparent, you maybe could compare the SR and CSCG on the original task of Shapiro et al. [6] or the current rooms experiment on a task version with and without aliased observations. For the original task version of Shapiro et al. [6] also a computed successor

representation could reveal communities as well as important bottleneck positions of the rooms/graph. For instance, Botvinick & Weinstein (2014) show that SR representations and abstractions are also essential to later apply higher-level/hierarchical learning/planning as shown with the CSCG in Fig. 7 d-h.

Since such hierarchical planning as in Fig. 7h is only possible due to the "posterior" community detection of the transition probabilities in Fig 7c/e). I wondered how easy such a procedure/mechanism could be added/implemented in the CSCG. And whether you could - again to emphasize the plausibility of the algorithm - speculate on how such high-level representations might emerge from lower-levels and also be represented on a neural level (regarding neural circuit).

Backup your claims & better understandability

It appears that in some sections not all claims nor statements are supported with sufficient evidence or backup. In general, please provide actual objective data/results or (more) references for statements made in this manuscript.

Add data / transparent comparison:

- Line 376: "making planning in the hierarchical CSCG-learned graph more efficient than [...]" -> more efficient is quite subjective could you please provide data or rephrase?

- Line 225: If possible, please provide how many observations/computation/random walks the CSCG needed to learn different graphs/rooms "perfectly". Same for other results in this manuscript (e.g., Line 175., Fig 2b.).

- Line 400: "CSCGs differs substantially from the Tolman-Eichenbaum Machine (TEM ...)" & Line 406: "TEM [...] requires a higher computational effort."

In this case, no objective comparison of the two models is presented. It is unclear if this comparison even results from two models performing the exact same task. The comparison of the "Lap Counting" task in Sun et al. 2020 with the task used in the TEM (BioRxiv) and this manuscript makes it questionable whether minor changes in the task details could also lead to computational differences as well as to different results (see also further criticism regarding the lap counting task).

To make a statement about any model comparison you should fit both models to the same tasks or data and provide objective evidence (in this case the code of the TEM is available). Also, please avoid to only pinpoint the "substantially differences" to one specific task/aspect of the model.

It even appears that this comparison solely results from own data on the CSCGs and a citation from the TEM paper. (CSCG: Line 407: "[...] uses 4 laps of 12 steps each, and is solved in seconds on a single CPU core [...]"; TEM: Line 408: "[...] with TEM, it needs to be simplified to 3 laps for 4 steps each" vs. original statement in the TEM manuscript: Page 18 "When TEM was trained on this task (using 3 laps instead of 4 for computational reasons), [...]" Whittington, et al. (2019) BioRxiv, 770495).

Overall, I would recommend avoiding such bold statements without enough objective evidence - even though this was/is part of the discussion section.

Elaborate for increased understandability

In some cases, statements such as "that make them biologically plausible" (Line 356) could be elaborated in the interest of the reader and "broader" audience of nature communications, rather than just listing one reference.

The same applies to the mentioned mechanism of "Replay" (Line 137, 452, 454, 455, and Fig. 1f and Fig. 4h). Since it is only mentioned briefly in the figure(s)/captions and text, the reader could benefit from a short elaboration/explanation of the mechanisms and concepts regarding (neural) replay.

Add references:

Line 386 "Current Theories of how cognitive maps are learned ..." → please add references or mention theories.

Minor Points

Overall, the manuscript itself and its clarity would benefit if some details/labels on the captions and figures could be added and typos/errors in the text and reference list could be corrected. In the interest of readability, I would recommend relying on coherent terminology and especially avoid misleading wording/terminology in some parts of the manuscript (see details below).

Figures

Figures and figure captions are mostly not self-explanatory, please also provide important details or labels that are mentioned in the text or methods sections but not in the figure captions. Therefore, the reader is left going back and forth through the manuscript. Some examples listed below:

Fig. 1) A) What is A, C, E, ... = Observations? What are the colored arrows? Blue Area = Context? Label for Specialization through context? C) Inconsistent with 1A and 1B, what is alpha, etc. D) What is the meaning of the different arrow colors? (see details text line 119); No labels for clones E) Color Labels: blue = observational bottom-up support; green = lateral contextual support); add details from the text line 132-134 F) Replay of what sequence?

Fig. 2) B) Could add a label for the CSCGs (similar to "first-order graph" in 2A) B&D) doubling of nodes, e.g., Fig. 2D. doubling of the yellow node in the right corner. Is this intentionally, remaining from the actual representation, or is it from the Viterbi training a remaining "redundant" clone? E) What is the meaning of the color gradient for observations? G) Color of both paths is hard to see H) Only label for "confounder" and "overlap", what is the rest? How many transitions or observations are shown by one dot of time? Does one dot represent one step/transition? How would the extracted transition Matrix look like (compared to Fig 7)?

Fig. 3) A) what is the observations color gradient? D) could add a label for "unaware of the obstacle", "discovered obstacle", "re-planning" E&F) could you quantify the improvement if the graph is known; "[...] can be used as a re-usable structure to quickly learn a new room with similar layout [...]" -> please rephrase that this is the same layout; also for the title "[...] explore similar environments", the caption for figure 3 G) is missing. Please add the detail from (Line 226), that the transition matrix was "kept fixed" between the different rooms, meaning the layout was "known"?!

Fig. 4) A) What is a, b, c, d, ... ? B) Could you explain why clones for the unaliased and not shared segment emerge (B, C, and M, N, ?) C) What is the meaning of the green and yellow color? D&E) add labels G) How is this learned transition graph efficient or higher-order as proposed in the introduction?

Fig. 5) A) Please mention that the start is unique between each lap; compared to Sun et al. (2020) the start is an "observation" that is always the same between all laps. Here, the task seems to be slightly optimized for the algorithm. B) What are the Colors? -> avoid the term "neurons" if only simulated! E) What are the different Colors? What are the Arrows (this should be understandable without the text or previous figures)?

Fig. 6) A&B) Red colors represent different observations? I) What is the y-axis? III) What are the green-colored areas? II – IV) "[...] produces global remapping, and [...] produces partial remapping" -> in the caption you state that CSCG produces remapping vs. in the text CSCG is the effect of remapping. Is this the cause or effect of remapping?

Fig. 7) G) Could you please quantify the efficiency of hierarchical planning.

Terminology

Please refer to simulated neural activity or a similar analogy instead of "neural activity", "neurons", "cells", "activations" → this could be misleading and appears incoherent over sections. I.e., Figure 5,

caption; Line 285 "weak activations"; Line 314 "neural responses"; Line 325 "neural response of two CSCGs"; Line 333 "neurons that fire in ..."; Line 777 and 795 "cells"; Line 250 and 781 "rat" → potentially meaning an agent?

Typos & smaller errors

Since it is common to develop some sort of "typo blindness" for the own manuscript, I would recommend finalizing it with proofreading again. Some errors are listed below:

- Line 149: instead of Fig 1d. you potentially meant Figure 1c.
- Line 60: "... similar to to neuro...."; Line 128: "... clones activate the the different ..."; Line 615: Springer with capital S; Line 647: "cloned" with capital C
- Line 483 & 563: Doubling of BioRxiv -Link

Reference list

Please update your reference list upon finalizing your manuscript, some of the BioRxiv citations are now already published in journals, for instance, reference 21 (Line 507).

Data and code availability

Line 857: Maybe you could also specify when your code will be available, for instance, upon publication, etc.

References

Aronov, D., Nevers, R., & Tank, D. W. (2017). Mapping of a non-spatial dimension by the hippocampal-entorhinal circuit. *Nature*, 543(7647), 719-722.

Botvinick, M., & Weinstein, A. (2014). Model-based hierarchical reinforcement learning and human action control. *Philosophical Transactions of the Royal Society B: Biological Sciences*, 369(1655), 20130480.

Niv, Y. (2019). Learning task-state representations. *Nature neuroscience*, 22(10), 1544-1553.

Schuck, N. W., & Niv, Y. (2019). Sequential replay of nonspatial task states in the human hippocampus. *Science*, 364(6447), eaaw5181.

Sun, C., Yang, W., Martin, J., & Tonegawa, S. (2020). Hippocampal neurons represent events as transferable units of experience. *Nature Neuroscience*, 23(5), 651-663.

Whittington, J. C., Muller, T. H., Mark, S., Chen, G., Barry, C., Burgess, N., & Behrens, T. E. (2019). The Tolman-Eichenbaum Machine: Unifying space and relational memory through generalisation in the hippocampal formation. *BioRxiv*, 770495.

REVIEWER COMMENTS

Reviewer #2 (Remarks to the Author):

Many thanks for responding to my previous points. I think this manuscript is much improved. I appreciate that you have had to deal with lots of other reviewer comments; however, some of my key suggestions still need to be addressed. Perhaps you could focus on the following three:

We thank the reviewer for pointing out these omissions. We have now added all the text suggested by the reviewer and we agree that those make the manuscript better.

1. Original recommendation: If you want to bring yourself up to speed with active inference in this setting, I would recommend ... (Kaplan and Friston, 2018) ... with many points at convergence with your own work. ... there is some pressure on you to consider the active inference literature in this area – and explain to readers where the different schemes sit in relation to each other.

Your response: We thank the reviewer for this very helpful suggestion. We have included several references that the reviewer suggests and juxtapose our model with active inference in the discussion section of our paper.

Very good. However, you do not cite or discuss Kaplan, R. and K. J. Friston (2018). "Planning and navigation as active inference." *Biol Cybern* 112(4): 323-343. Can I suggest you look at this paper and replace the following sentence in your discussion:

"Although active inference has been so far used in far simpler models that would not be able to solve the experiments presented in the current work, CSCG's probabilistic formulation is compatible with representing the certainty of the model using hierarchical priors over model parameters, offering an avenue for future research."

with

"Active inference and CSCG's adopt a probabilistic formulation that accommodates uncertainty using hierarchical priors over model parameters. This offers an avenue for further research into structure learning and planning as inference, in this setting. For example, please see (Kaplan & Friston, 2018) for an application of active inference to spatial planning, navigation, and path cells."

Thank you. We have now included this. (Lines 491-495)

2. Original recommendation: I think it would be useful to include the following here: "Intuitively, each step of the EM algorithm updates posterior beliefs about hidden states and corresponds to state estimation or inference. Conversely, the M step updates point estimators of model parameters and can be construed as learning. As with the Tolman Eichenbaum machine, these expectation maximisation-based schemes (e.g., the Baum-Welsh algorithm) effectively ignore uncertainty about the parameters and replace posteriors over parameters with point estimates or Delta functions. This has the computational benefit of not having to worry about conditional dependencies between posterior densities over states and parameters."

Your response: We thank the reviewer for this suggestion and have included this paragraph in the manuscript.

You have not included this paragraph. Please include it. Although technical, this will be useful for people who understand the maths.

We have now included this. (Lines 476-482)

3. Original recommendation: you could then explain why this differs from active inference with something like:

"Conversely, in active inference approximate posteriors are updated over both states and parameters. This means that uncertainty about parameters nuances estimates of hidden states and vice versa. Technically, active inference schemes, in this setting, generally use some form of variational Bayes under a mean field approximation (of which the EM algorithm can be seen as a special case). The mean field approximation for the parameters of the transmission and emission matrices are generally parameterised in terms of Dirichlet distributions, which leads to simple update schemes that include both the uncertainty about model parameters and their expected values. Although not considered here, including uncertainty about parameters can be important in terms of optimising exploration or searches – to minimise uncertainty about emission (i.e., likelihood) and transmission (i.e., prior) probabilities."

Your response: We thank the reviewer for this very helpful suggestion. We have included this paragraph in the discussion section of our revised manuscript.

You have not included this paragraph. Please include in it. Although technical, this will be useful for people who understand the maths.

We have now included the paragraph. (Lines 483-495)

Thank you for these recommendations that helped to significantly improve the discussion about the model.

Reviewer #4 (Remarks to the Author):

Review: „Learning cognitive maps as structured graphs for vicarious evaluation“

Summary

The manuscript discusses on a mechanistic level how cognitive maps could be represented and how this framework possibly could apply to cell activities of the hippocampal formation. The presented algorithm enables multiple representations (“clones”) of identical sensory observations in different environments/contexts while maintaining exact spatial/relational information. Interestingly, such a “cloned” representation then allows for behavioral responses specific to a current context. The authors also propose a neurobiological-circuit on how these cloned observations could be represented on a cellular level, for instance, how recruited cell activity on a population level could signal the context of a given environment and handle uncertainty. The link to (cognitive) neuroscience is drawn through a vast number of classical neurophysiological experiments that are adapted to be solved by the proposed algorithm. Beside some interesting similarities between the model-driven results, the algorithm also would allow investigating further questions.

In sum this is an exciting manuscript using a novel computational approach. However, (biological) plausibility of the model is not clear to me (see below) and the correction of some formal errors would significantly improve the manuscript and convey the main idea more clearly.

Thank you for your insightful comments on the manuscript. We are happy that you found the manuscript exciting, and we have used your feedback to improve it further. The changes are highlighted in the diff file we are uploading with this manuscript.

We want to start by addressing one of the points you raise later in the review because this might affect the overall perception about the paper. The relevant section from the review is quoted below:

“In the methods section, the start and stop for each lap of Fig. 5a task is described as unique (Line 789: “The start and end positions were unique observations.”). Could you please clarify if these start and end positions were unique for each lap vs. only for the first and last lap of the task. If yes, this manipulation seems unfortunately not comparable to the original study by Sun et al. (2020). Adding a unique observation to each start of a lap is changing the initial task substantially. Therefore, the observed results are less surprising and less comparable to the cellular response reported by Sun et al. (2020).”

We are happy to report that we do not use a unique start/stop marker for each lap. CSCG is learning to unwrap the repetition of the sequences into different clones on its own because it helps to predict the reward at the end of the 4th lap. I.e. the training sequence is “Start, 1, 2, 3, ...,12, 1, 2, 3, ..., 12, 1, 2, 3, ..., 12, 1, 2, 3,, 12, Reward”. We agree with you that adding a marker for each lap would defeat the purpose, and wouldn’t be compelling.

We have modified the manuscript to remove this ambiguity by mentioning this clearly in the caption for Fig 5.

Major Points

Dimensionality / Higher-Order Graph / Complexity Reduction

Reduction of dimensionality or compression of low-level representations is useful when you want to use/represent data more efficiently - this seems especially plausible when an abstraction of latent state spaces or environments enables fast and simple inference or computations. Therefore, such detailed action-observation representation also containing plenty of redundant information (see Figure 2c and d) might not be the most (biologically) plausible nor efficient form. Yet it enables extracting all exact locations in an environment, even if observations are aliased. The authors explain how a first-order graph would “under-fit the sequence/data”. How does this compare to the CSCG? It appears that in contrast, the CSCG is partly over-fitting a sequence by combining even redundant/aliased observations with unique actions (Fig. 2d), resulting in this 1:1 representation with all exact spatial locations/relations. Could such representation also be learned without any action-observation pairs only resulting from a sequence/stream of sensory observations without actions or knowledge about relations that caused each transition?

CSCG is not overfitting the data because even memorizing the whole training sequence will not lead to correct predictions in the room examples -- a new random walk would still throw the model off. Instead, CSCG can be viewed as recovering the generative structure that generated the data. This is why it is able to recover the room layout, routes, or laps depending on the sequential observations. Note also that the combination of observations+unique actions is not sufficient to uniquely identify the locations in each of the cases we have shown. As an example, a sequence like “red (right) blue (left) red” could be interpreted by our model as having 3 distinct locations, with the first and the last red being different (which would not match the actual topology). However, the right topology provides a simpler explanation, particularly for long sequences, and thus it is favored by CSCG.

Learning without observing actions is an interesting case, and we have now included this in the revision as part of supplementary results. Without observing actions, the model can still recover room layouts, but the recovery for our original room size is imperfect compared to the case in which actions are available. Smaller room sizes result in perfect recovery.

- In the supplement we added an experiment on learning without observing actions
- We refer to this experiment in the main text.

High Dimensionality

The authors should comment on how their model but also the proposed neural circuit could represent multidimensional environments and potentially extract only what is salient and/or relevant at different settings. For instance, stimuli/observations containing information from multiple dimensions, where only two dimensions are relevant to solve a task. Or somewhat more generally spoken, experiences in the real life are never exactly the same, therefore the question of how the model would differentiate relevant and irrelevant information and could represent only the most necessary/abstract information.

Dealing with multi-dimensional environments can be accomplished by having a clustering or co-occurrence detection step before CSCG. The set of co-occurrences (or cluster-centers) become the alphabet over which CSCG learns the sequences. Note that this is consistent with the view expressed in Buzsaki & Tingley and others (see figure below)

Trends in Cognitive Sciences

Figure 4. Hippocampal Sequencing Hypothesis. Indices that point to cortical, and subcortical, modules for different inputs are sequenced by hippocampal activity patterns, thus preserving the ordinal structure over which experience occurs. Abbreviations: Aud, auditory; Olf, olfactory; Som, somatosensory; Vis, visual.

CSCG is purely dealing with the sequencing aspect of the problem, where each observation is treated as a nominal cluster center index. But the cluster itself can be a multi-dimensional co-occurrence. Although the clustering step itself is not addressed in our work, prior work (Mok & Love 2019 for example) has explored this part of the problem. CSCG will treat imperfect matches to cluster centers as soft evidence.

CSCG and clustering step could potentially be combined into a unified graphical model. We hope to work on a combined graphical model that includes this clustering step and CSCG as part of future research. Another direction for including multidimensionality is to have factorial CSCGs analogous to factorial HMMs. This is also another line of investigation we hope to take up in the future.

We have added these points as part of the Discussion section

(Neuro-)biological plausibility

The proposed neural circuit idea is quite compelling and interesting. As proposed, the output of neurons (for one observation) is weighted by the sum of its lateral inputs but also multiplied by the bottom-up input (Line 123). This could also have some great applications regarding predictive coding. As described in Line 139 the “[...] population of clone neurons represent the probability of different contexts [...]” this mechanism could also be quite convincing on how to handle noisy environments/input. And how “soft evidence” in case of uncertainty could be passed-on through lateral inputs.

But a few questions remain unclear. Could the authors comment on how such single neuron representations might emerge in the first place? And additionally, argue how efficient this one neuron one representation/clone mechanism might be. Potentially regarding other cells such as place cells, where single neurons represent/respond to multiple environments (see also Remapping). Or even the same place and grid cells reportedly represent both spatial and non-spatial information also across sensory modalities (Aronov et al. 2017). In contrast, in their proposed neural circuit one single cell only would respond to one observation from one specific context, if aliased this would result in another cellular representation. This might lead to a large quantity of neurons representing complex/high-dimensional environments or non-spatial dimensions.

1) The clone-based representation itself is efficient and sparse in comparison to models that represent context by accumulating history of observed samples. As an example, consider the task of storing ngrams as in a language model. The clone-based representation will use much fewer neurons compared to ngrams because the ngrams will strictly need to keep one clone for every single context it has encountered. CSCGs will merge clones that follow the same paths, and collapse clones appropriately when a first order, or zeroth order graph is sufficient for part of the data. This flexibility in the graph representation makes efficient use of the cloned structure. The model doesn't use a clone per single context -- it uses a clone per flexible context.

This can be seen in the case of the rooms experiment (Fig 2A). If we store 10-grams of sequences observed in room1 -- which will correspond to having a clone per strict context -- we need on the order of a million (4^{10}) clones. But the model actually uses only ~10 clones per observation

As we described earlier, the sequencing model is assumed to sit above a co-occurrence clustering model. The mixing of spatial and non-spatial information is assumed to be happening in this co-occurrence clustering stage, and the CSCG then deals with sequencing those clusters.

Moreover, when different environments have similar parts, the CSCG can share those similar parts, and this often happens. In addition, hierarchy formation is another way in which CSCG deals with representational efficiency.

Context

Regarding the context (encoded from the weighted lateral input of neurons, Line 131), I wondered whether the authors could specify what comes first: Single-cells that encode a current context vs. a novel context/environment that leads to aliased and new single-cell representations? Maybe the authors could speculate what other structures and input would be needed to form novel single-cell representations of aliased observations or a novel environment/context. For instance, it is suggested that the hippocampus determines when to create a new state or update an old one (see Niv 2019), and vice versa, the context is needed to detect novelty. Moving away from the mechanistic approach/single-cell representations, the authors should also comment how different brain areas might be relevant for the acquisition and representation of different cognitive maps, e.g., abstract task states that are represented in the OFC (Schuck & Niv 2019).

In the current implementation of CSCG, the single cells will initially be primarily driven by the immediate observation, and with experience the clones will learn to differentiate between the different environmental contexts in which that observation occurs. If there aren't enough clones to model different contexts, CSCG would generalize learning widely, as in literature cited in Niv'19. While our current work offers learning and inference mechanisms and a biological circuit, committing these mechanisms to different brain regions is left for future work. Hippocampus is known to play important roles in creating new states to represent new contexts, in updating older ones, and in detecting novelty -- suggesting that the core mechanisms of CSCG could be implemented in the hippocampus, potentially in CA3 or CA1. Decision making based on a learned transition graph depends on evaluating the value of different options under uncertainty,

and under a task specification, and the orbito-frontal cortex could be the locus of those computations.

We have added the above points to the Discussion section.

Known Graph Structure (Figure 3)

Since the graph/room layout stays exactly the same between relearning of a novel arrangement of observations (room 1 vs. room 2, I wondered how the re-learning performance of only partly overlapping or even incongruent layouts with aliased observations would look like – if applicable please provide data on that.

Relearning a smaller room will work just as well because it will just map to a subgraph of the original graph. Learning incongruent layouts that are larger than the original layouts will produce a gradual degradation in performance with increase in the size mismatch. We have added this result to the supplement.

- We performed additional experiments to explore how larger rooms will be represented using the transition-graph schema of a smaller reference room. These results are added in the supplement
- We refer to these supplementary results in the main text.
- We reworded the corresponding results section to emphasize that the results shown in the main paper are for the exact match of layout and that when the rooms do not exactly match in layout there is a degradation in performance.

Lap Counting Task (Figure 5)

In the methods section, the start and stop for each lap of Fig. 5a task is described as unique (Line 789: “The start and end positions were unique observations.”). Could you please clarify if these start and end positions were unique for each lap vs. only for the first and last lap of the task. If yes, this manipulation seems unfortunately not comparable to the original study by Sun et al. (2020). Adding a unique observation to each start of a lap is changing the initial task substantially. Therefore, the observed results are less surprising and less comparable to the cellular response reported by Sun et al. (2020).

A unique observation between the laps is clearly allowing the CSCG to differentiate the different laps and even be able to count the unique observations (laps) until a specific goal is reached (4 laps). In contrast, in the original paper by Sun et al. (2020), all observations were the same for the rodents navigating a maze for 4 laps until a reward was received. Albeit all laps were exactly the same, yet the cellular pattern showed this distinctive result included in figure 5c. Since this task and results are quite important for their manuscript in explaining the plausibility and also capabilities of the proposed model, the authors should clarify this task detail and your findings regarding this slight but important change of the task manipulation.

We addressed this earlier, repeating it here for completeness. We are happy to report that we do not use a unique start/stop marker for each lap. CSCG is learning to unwrap the repetition of the sequences into different clones on its own because it helps to predict the reward at the end of the 4th lap. I.e. the training sequence is “Start, 1, 2, 3, ..., 12, 1, 2, 3, ..., 12, 1, 2, 3, ..., 12, 1, 2, 3, ..., 12, Reward”. We agree with you that adding a marker for each lap would defeat the purpose, and won’t be compelling.

We have modified the manuscript to remove this ambiguity by mentioning this clearly in the caption for Fig 5.

Additionally, one of the many manipulations in Sun et al. (2020) was to elongate the trained maze, as similarly proposed in this manuscript (Figure caption 5.d. – unfortunately not part of the methods section). Here I would also argue against the comparability of both tasks. In your manuscript „[...] 4 laps of 12 steps each [...]“(Line 407) were elongated but with a fixed/the same amount of observations (i.e., 12). How would the event-specific representation in Fig. 5.d. change, if also the number of observations would be increased (elongated maze -> more observations)? Furthermore, it would be interesting to see, how these representations would change through a different shape of the maze or after training on 4 laps but showing the reward (unique observation/goal) after 5 laps, etc. See other manipulations in Sun et al. (2020).

When a maze is physically elongated, the segments of the maze can be considered as repeating. In that sense, we believe that repeating the observations corresponds to a physical elongation of the maze, and that is why we chose repeating observations for the elongated maze.

Even if elongation of the maze creates entirely new observations in the elongated space, CSCG is able to recover due to smoothing and probabilistic inference. We have added these results to the supplement.

- We did an additional experiment to test whether clone activation traces would still be preserved when the maze is elongated using novel observations. The answer is yes.
- We have added this result to the supplement, and we refer to it from the main paper.

Remapping (Figure 6)

Regarding re-mapping (Fig 6.), I would recommend highlighting, that only one CSCG was trained on all different rooms (and one for the maze set). Since all rooms were aliased this results in representing different rooms with different clones for each observation. Here it is interesting to see, how a different amount of training or noise results in differences in the beliefs and identifying the rooms. But in respect of the proposed neural circuit (biological plausibility) cloned observations (of different rooms) would be represented by different neurons and this, in fact, is not entirely in accordance with the common assumption of re-mapping of e.g., place cell activity. Also, please be considerate with the “cause or result” of remapping, whether CSCG produces remapping or could explain remapping. Please also check for inconsistencies in the text and figure captions, since this could be misleading.

We have edited the section to highlight that only one CSCG is trained for each environment set, and fixed the inconsistencies in some portions of the text.

As we explain, whether different environments will have different clones or not will depend on the amount of experience and the similarity between the environments. Highly similar environments can have some segments that are overlapping (represented by the same clones) even after prolonged training.

- We edited the section thoroughly to avoid inconsistencies, and to highlight the points you asked for.

Hierarchical Abstraction (Figure 7)

As reported in Fig 7 a-b) the SR fails to reveal modularity in the given task with aliased observations. To be fair or more transparent, you maybe could compare the SR and CSCG on the original task of Shapiro et al. [6] or the current rooms experiment on a task version with and without aliased observations. For the original task version of Shapiro et al. [6] also a computed successor representation could reveal communities as well as important bottleneck positions of the rooms/graph. For instance, Botvinick & Weinstein (2014) show that SR representations and abstractions are also essential to later apply higher-level/hierarchical learning/planning as shown with the CSCG in Fig. 7 d-h.

We believe this is a misconception, partially due to the way results are presented in Botvinick & Weinstein 2014. Schapiro et al shows a neural network that can discover communities, and Botvinick & Weinstein show that similarity metric on SR can discover clusters. **They do not show that community detection on the transition graph (the first order Markov chain) cannot discover the same communities.**

We ran CSCG on the original graph in Schapiro et al, and Botvinick & Weinstein 2014. As expected, CSCG learns the graph because the learning problem is simpler for the fully-observed setting. A community detection on the learned CSCG successfully identifies the clusters. **This shows that forming an SR is not required for discovering the communities -- just the transition graph is enough.**

- We have included this result in the supplement, and we reference it from the main paper.

Since such hierarchical planning as in Fig. 7h is only possible due to the “posterior” community detection of the transition probabilities in Fig 7c/e). I wondered how easy such a procedure/mechanism could be added/implemented in the CSCG. And whether you could - again to emphasize the plausibility of the algorithm - speculate on how such high-level representations might emerge from lower-levels and also be represented on a neural level (regarding neural circuit).

We speculate that community detection and learning can be treated together as a mixture of CSCGs. We have included this in the discussion section, and exploring the biological plausibility of this setup is left as future exploration. The fact that Belief Propagation -- a local message-passing algorithm -- is one of the best methods for solving community detection gives us hope that a biologically plausible method can be found.

- We have added these to the main text in the Results section and in the Discussion section.

Backup your claims & better understandability

It appears that in some sections not all claims nor statements are supported with sufficient evidence or backup. In general, please provide actual objective data/results or (more) references for statements made in this manuscript. Add data / transparent comparison:

- Line 376: "making planning in the hierarchical CSCG-learned graph more efficient than [...]" -> more efficient is quite subjective could you please provide data or rephrase?

The original manuscript included a paragraph on the efficiency of hierarchical planning in the supplement (line 838 in the original), but we can see why it could be missed. To call attention to this, we have added the following in the appropriate place in the manuscript.

“(See Supplementary Methods for more implementation details and computational efficiency estimates)”

- Line 225: If possible, please provide how many observations/computation/random walks the CSCG needed to learn different graphs/rooms "perfectly". Same for other results in this manuscript (e.g., Line 175., Fig 2b).

The original manuscript included these details in the supplement, we have now included the length of the random walk in the main text for these experiments.

- Line 400: "CSCGs differs substantially from the Tolman-Eichenbaum Machine (TEM ...)" & Line 406: "TEM [...] requires a higher computational effort."

In this case, no objective comparison of the two models is presented. It is unclear if this comparison even results from two models performing the exact same task. The comparison of the "Lap Counting" task in Sun et al. 2020 with the task used in the TEM (BioRxiv) and this manuscript makes it questionable whether minor changes in the task details could also lead to computational differences as well as to different results (see also further criticism regarding the lap counting task).

To make a statement about any model comparison you should fit both models to the same tasks or data and provide objective evidence (in this case the code of the TEM is available). Also, please avoid to only pinpoint the "substantially differences" to one specific task/aspect of the model.

It even appears that this comparison solely results from own data on the CSCGs and a citation from the TEM paper. (CSCG: Line 407: "[...] uses 4 laps of 12 steps each, and is solved in seconds on a single CPU core [...]"; TEM: Line 408: "[...] with TEM, it needs to be simplified to 3 laps for 4 steps each" vs. original statement in the TEM manuscript: Page 18 "When TEM was trained on this task (using 3 laps instead of 4 for computational reasons), [...]" Whittington, et al. (2019) BioRxiv, 770495).

Overall, I would recommend avoiding such bold statements without enough objective evidence - even though this was/is part of the discussion section.

Thank you for pointing this out. We appreciate the reviewer's concern here, and we have substantially changed the wording in this section.

- We rewrote this paragraph in the Discussion section to address these points.

Elaborate for increased understandability

In some cases, statements such as "that make them biologically plausible" (Line 356) could be elaborated in the interest of the reader and "broader" audience of nature communications, rather than just listing one reference.

We have included additional context in Figures and the main text to address this concern raised by the reviewer.

The same applies to the mentioned mechanism of "Replay" (Line 137, 452, 454, 455, and Fig. 1f and Fig. 4h). Since it is only mentioned briefly in the figure(s)/captions and text, the reader could benefit from a short elaboration/explanation of the mechanisms and concepts regarding (neural) replay.

We have added this

Add references:

Line 386 "Current Theories of how cognitive maps are learned ..." → please add references or mention theories.

We have added these

Minor Points

Overall, the manuscript itself and its clarity would benefit if some details/labels on the captions and figures could be added and typos/errors in the text and reference list could be corrected. In the interest of readability, I would recommend relying on coherent terminology and especially avoid misleading wording/terminology in some parts of the manuscript (see details below).

Figures

Figures and figure captions are mostly not self-explanatory, please also provide important details or labels that are mentioned in the text or methods sections but not in the figure captions. Therefore, the reader is left going back and forth through the manuscript. Some examples listed below:

Fig. 1) A) What is A, C, E, ... = Observations? What are the colored arrows? Blue Area = Context? Label for Specialization through context? C) Inconsistent with 1A and 1B, what is alpha, etc. D) What is the meaning of the different arrow colors? (see details text line 119); No labels for clones E) Color Labels: blue = observational bottom-up support; green = lateral contextual support); add details from the text line 132-134 F) Replay of what sequence?

We have modified this figure and its caption:

- 1) Added the labels as the reviewer suggested
- 2) Modified the caption to include more details

Fig. 2) B) Could add a label for the CSCGs (similar to "first-order graph" in 2A) B&D) doubling of nodes, e.g., Fig. 2D. doubling of the yellow node in the right corner. Is this intentionally, remaining from the actual representation, or is it from the Viterbi training a remaining "redundant" clone? E) What is the meaning of the color gradient for observations? G) Color of both paths is hard to see H) Only label for "confounder" and "overlap", what is the rest? How many transitions or observations are shown by one dot of time? Does one dot represent one step/transition? How would the extracted transition Matrix look like (compared to Fig 7)?

We have modified this figure and its caption as suggested by the reviewer:

- Added labels within the figure for clarity
- Removed the confusing colorbar
- Expanded the caption to include more details
- Changed the colors of the paths in g for more clarity.

Fig. 3) A) what is the observations color gradient? D) could add a label for "unaware of the obstacle", "discovered obstacle", "re-planning" E&F) could you quantify the improvement if the graph is known; "[...]" can be used as a re-usable structure to quickly learn a new room with similar layout "[...]" → please rephrase that this is the same layout; also for the title "[...] explore similar environments", the caption for figure 3 G) is missing. Please add the detail from (Line 226), that the transition matrix was "kept fixed" between the different rooms, meaning the layout was "known"?!

We have made the following changes as suggested by the reviewer:

- Added labels to the figure
- Removed the confusing colorbar

- Edited the caption to make it clear that the graphs show re-learning an identical hidden layout.

Fig. 4) A) What is a, b, c, d, ... ? B) Could you explain why clones for the unaliased and not shared segment emerge (B, C, and M, N, ?) C) What is the meaning of the green and yellow color? D&E) add labels G) How is this learned transition graph efficient or higher-order as proposed in the introduction?

We have made the following changes based on suggestions from the reviewer:

- Added labels for clarity
- Added colorbar to Fig c.
- Added more details to the captions for clarity.

Fig. 5) A) Please mention that the start is unique between each lap; compared to Sun et al. (2020) the start is an "observation" that is always the same between all laps. Here, the task seems to be slightly optimized for the algorithm. B) What are the Colors? -> avoid the term "neurons" if only simulated! E) What are the different Colors? What are the Arrows (this should be understandable without the text or previous figures)?

We have made the following changes as suggested by the reviewer:

- Emphasized the point that there is no unique start state for each lap to help avoid confusion.
- Added colorbar
- Added details to the caption to explain the meaning of colors in e.

Fig. 6) A&B) Red colors represent different observations? I) What is the y-axis? III) What are the green-colored areas? II – IV) "[...] produces global remapping, and [...] produces partial remapping" -> in the caption you state that CSCG produces remapping vs. in the text CSCG is the effect of remapping. Is this the cause or effect of remapping?

We have made the following changes as suggested by the reviewer:

- In the caption we mention that the different colors represent different observations
- Added y axis labels and plot titles.
- Removed the confusing green bars
- Edited text for clarity.

Fig. 7) G) Could you please quantify the efficiency of hierarchical planning.

We have included this in the supplement, and we refer to it from the main paper.

Terminology

Please refer to simulated neural activity or a similar analogy instead of "neural activity", "neurons", "cells", "activations" → this could be misleading and appears incoherent over sections. I.e., Figure 5, caption; Line 285 "weak activations"; Line 314 "neural responses"; Line 325 "neural response of two CSCGs"; Line 333 "neurons that fire in ..."; Line 777 and 795 "cells"; Line 250 and 781 "rat" → potentially meaning an agent?

Thank you for the detailed notes. We have cleaned up the terminology at these places, and throughout the manuscript.

Typos & smaller errors

Since it is common to develop some sort of "typo blindness" for the own manuscript, I would recommend finalizing it with proofreading again. Some errors are listed below:

- Line 149: instead of Fig 1d. you potentially meant Figure 1c.
- Line 60: "... similar to to neuro..."; Line 128: "... clones activate the the different ...";

Thank you. We have corrected these.

Line 615: Springer with capital S; Line 647: "cloned" with capital C
- Line 483 & 563: Doubling of BioRxiv -Link

Reference list

Please update your reference list upon finalizing your manuscript, some of the BioRxiv citations are now already published in journals, for instance, reference 21 (Line 507).

Data and code availability

Line 857: Maybe you could also specify when your code will be available, for instance, upon publication, etc.

We will make the code available at the time of publication, or hopefully even before that. A few labs are using our code already, we just wanted to get this revision completed before taking up the task of making the code ready for release.

We have made this change in the manuscript, and updated the references.

References

Aronov, D., Nevers, R., & Tank, D. W. (2017). Mapping of a non-spatial dimension by the hippocampal–entorhinal circuit. *Nature*, 543(7647), 719-722.

Botvinick, M., & Weinstein, A. (2014). Model-based hierarchical reinforcement learning and human action control. *Philosophical Transactions of the Royal Society B: Biological Sciences*, 369(1655), 20130480.

Niv, Y. (2019). Learning task-state representations. *Nature neuroscience*, 22(10), 1544-1553.

Schuck, N. W., & Niv, Y. (2019). Sequential replay of nonspatial task states in the human hippocampus. *Science*, 364(6447), eaaw5181.

Sun, C., Yang, W., Martin, J., & Tonegawa, S. (2020). Hippocampal neurons represent events as transferable units of experience. *Nature Neuroscience*, 23(5), 651-663.

Whittington, J. C., Muller, T. H., Mark, S., Chen, G., Barry, C., Burgess, N., & Behrens, T. E. (2019). The Tolman-Eichenbaum Machine: Unifying space and relational memory through generalisation in the hippocampal formation. *BioRxiv*, 770495.

** See Nature Research's author and referees' website at www.nature.com/authors for information about policies, services and author benefits.

COVID 19 and impact on peer review

As a result of the significant disruption that is being caused by the COVID-19 pandemic we are very aware that many researchers will have difficulty in meeting the timelines associated with our peer review process during normal times. Please do let us know if you need additional time. Our systems will continue to remind you of the original timelines but we intend to be highly flexible at this time.

This email has been sent through the Springer Nature Tracking System NY-610A-NPG&MTS

Confidentiality Statement:

This e-mail is confidential and subject to copyright. Any unauthorised use or disclosure of its contents is prohibited. If you have received this email in error please notify our Manuscript Tracking System Helpdesk team at <http://platformsupport.nature.com> .

Details of the confidentiality and pre-publicity policy may be found here <http://www.nature.com/authors/policies/confidentiality.html>

Privacy Policy | Update Profile

DISCLAIMER: This e-mail is confidential and should not be used by anyone who is not the original intended recipient. If you have received this e-mail in error please inform the sender and delete it from your mailbox or any other storage mechanism. Springer Nature America, Inc. does not accept liability for any statements made which are clearly the sender's own and not expressly made on behalf of Springer Nature America, Inc. or one of their agents.

Please note that neither Springer Nature America, Inc. or any of its agents accept any responsibility for viruses that may be contained in this e-mail or its attachments and it is your responsibility to scan the e-mail and attachments (if any).

Reviewers' Comments:

Reviewer #4:

Remarks to the Author:

The major revisions of the authors significantly improved the ms and increased clarity and comprehensibility. It is great to see how CSCGs could be implemented in a large number of adapted experiments that now (partly) gain additional significance through further supplemental figures/experiments, but also through important details that are clarified and highlighted more prominently in the text (e.g., the lap counting exp.).

However, I would like to emphasize that both the introduction and the discussion still require revision concerning length, structure, and readability of the text.

Regarding the discussion, the mentioned topics/paragraphs cover many relevant areas, but ideally, the authors could reconsider which are important for the main message of the manuscript and link back to the results and introduction. So far, after a brief summary of the results, these are followed by several paragraphs referring to relevant literature or other models, etc., that unfortunately seem to be somewhat loosely arranged.

Obviously, this might be challenging since all points in the discussion have their justification, but they could be initiated or arranged in a more structured and logical way. Here, the readability may have also been somewhat reduced by suggested changes throughout the long review process. I would recommend revising the structure and rethink content that is important to get the "story" across.

Having a focus on (a) key statement(s) and logical as well as structured backup and literature integration for these points would definitely support the overall statement in lines 496 - 502 and the presentation of the manuscript, also in the interest of the reader and the broad audience of Nature Communications.

Response to Reviewer 4

Reviewer #4 (Remarks to the Author):

The major revisions of the authors significantly improved the ms and increased clarity and comprehensibility. It is great to see how CSCGs could be implemented in a large number of adapted experiments that now (partly) gain additional significance through further supplemental figures/experiments, but also through important details that are clarified and highlighted more prominently in the text (e.g., the lap counting exp.).

However, I would like to emphasize that both the introduction and the discussion still require revision concerning length, structure, and readability of the text.

Regarding the discussion, the mentioned topics/paragraphs cover many relevant areas, but ideally, the authors could reconsider which are important for the main message of the manuscript and link back to the results and introduction. So far, after a brief summary of the results, these are followed by several paragraphs referring to relevant literature or other models, etc., that unfortunately seem to be somewhat loosely arranged.

Obviously, this might be challenging since all points in the discussion have their justification, but they could be initiated or arranged in a more structured and logical way. Here, the readability may have also been somewhat reduced by suggested changes throughout the long review process. I would recommend revising the structure and rethink content that is important to get the “story” across.

Having a focus on (a) key statement(s) and logical as well as structured backup and literature integration for these points would definitely support the overall statement in lines 496 - 502 and the presentation of the manuscript, also in the interest of the reader and the broad audience of Nature Communications.

Our Response:

We thank the reviewer for the comments. We are happy to learn that the reviewer found our modifications to be adding significance and clarity to the manuscript.

As the reviewer suggested, we have streamlined and shortened the Discussion section, while remaining consistent with Reviewer-2's comments as well. The improved Discussion section is ~25% shorter than the previous one, and is organized more coherently compared to the previous version. We also made edits in the introduction section for clarity.